# Multifaceted consequences of visual distraction during natural behaviour
Levi Kumle [1,2] ✉, Melissa L.-H. Võ [3], Anna C. Nobre [1,2,4] & Dejan Draschkow[1,2]

Visual distraction is a ubiquitous aspect of everyday life. Studying the consequences of distraction during temporally extended tasks, however, is not tractable with traditional methods. Here we developed a virtual reality approach that segments complex behaviour into cognitive subcomponents, including encoding, visual search, working memory usage, and decision-making. Participants copied a model display by selecting objects from a resource pool and placing them into a workspace. By manipulating the distractibility of objects in the resource pool, we discovered interfering effects of distraction across the different cognitive subcomponents. We successfully traced the consequences of distraction all the way from overall task performance to the decision-making processes that gate memory usage. Distraction slowed down behaviour and increased costly body movements. Critically, distraction increased encoding demands, slowed visual search, and decreased reliance on working memory. Our findings illustrate that the effects of visual distraction during natural behaviour can be rather focal but nevertheless have cascading consequences.

Consider following a recipe when baking. We have no trouble finding the necessary utensils and ingredients in the kitchen and combining them into a comforting product. Even as children, we can follow Lego instructions, finding the required pieces and assembling them in the right order into our creations. During such natural behaviours, we often encounter many competing visual objects (distractors) while we hold the relevant object in mind (e.g., seeing the flour as we search for the sugar or rummaging through many Lego bricks to find the red tile). Despite the distractions, we usually succeed in completing our behavioural goals.

Effects of distraction on core components of natural behaviour, such as working memory (WM)[1–10], attentional allocation during visual search[11–22], and decision-making[23–26] have long been investigated in separate branches of research. This separation has been essential for addressing fundamental questions about the mechanisms that help us handle distracting information during perceptual tasks (reviewed in refs. 27–33) and protect information we hold in in mind from distraction (reviewed in refs. 34,35).

However, accomplishing complex goals during natural behaviour often necessitates the coordination of multiple cognitive processes. Understanding distraction during unconstrained behaviours, therefore, requires considering cognitive processes – and their interconnections– jointly[36–40]. For instance, imagine it takes us longer to find the sugar while we are assembling the recipe because the kitchen is very cluttered. Distraction, in this example, interferes with an isolated cognitive subcomponent: visual

search. However, would such interference affect other cognitive subcomponents once we found the sugar? Would we structure our overall behaviour differently, knowing the kitchen is cluttered? We know little about the temporally extended consequences of distraction during naturally unfolding cognition.

Here, we developed a virtual reality (VR) approach to separate and quantify core cognitive subcomponents in an immersive context to study the impact of visual distraction in naturally unfolding behaviour. During an adapted object-copying task[41,42], participants copied a model display by selecting realistic objects from a resource pool and placing them into a workspace (Fig. 1a, Supplementary Movie 1). Compared to traditional laboratory studies, participants could freely decide when to look back at or move away from the model display (i.e., stop and start encoding). That is, participants could choose between using their memory representations to guide behaviour (memory-guided behaviour) and looking back at objects within the model.

Building upon a rich theoretical tradition of breaking down complex behaviour into subtasks[36,37], our VR method and the sequential nature of the task allowed us to segment continuous behaviour into elemental cognitive subcomponents (Fig. 1b). We tracked head, hand, and eye movements as well as their interactions with the virtual environment to quantify subcomponents such as encoding, visual search, and working-memory usage. Critically, this approach allowed us to capture sensorimnemonic decisions (i.e., decisions about whether to rely on information in the external

[1]Department of Experimental Psychology, University of Oxford, Oxford, UK. [2]Wellcome Centre for Integrative Neuroimaging, University of Oxford, Oxford, UK. [3]Department of Psychology, Goethe University Frankfurt, Frankfurt, Germany. [4]Wu Tsai Institute and Department of Psychology, Yale University, New Haven, CT, USA. ✉e-mail: levi.kumle@psy.ox.ac.uk

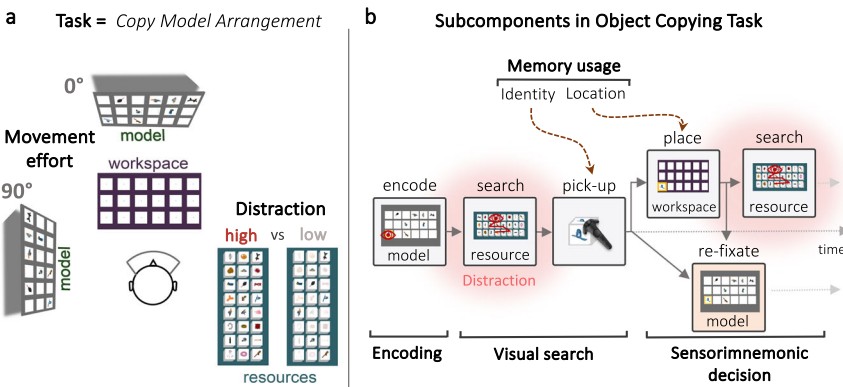

**Fig. 1 | Subcomponents in VR Object copying task. a** In the temporally extended VR protocol, participants copied a "model" arrangement by selecting realistic objects from a "resource" pool and placing them into a "workspace" (Supplementary Movie 1). By manipulating the transparency of the distractor objects in the resource pool, the participants completed the task in either a high or low distraction condition (Supplementary Movie 2). When copying the objects, the model was positioned either directly above the workspace (low movement effort) or rotated 90° (high movement effort, Supplementary Movie 3). **b** Our VR protocol allowed us to break down the overall behavioural goal of copying the model arrangement by capturing several proxy measures of hallmark cognitive subcomponents. First, participants had to encode information from the model (i.e., Encoding). Next, participants searched for encoded objects in the resource pool (i.e., Visual search). Both picking up and placing an object additionally required participants to use information stored in memory (i.e., Memory usage). Critically, participants could choose when to look back at the model (rely on the external environment) or when to rely on memory to guide behaviour (i.e., Sensorimnemonic decisions).

environment versus on information stored in memory; Fig. 1b), which have received limited scientific attention.

Critically, participants completed the task in either a high- or low-distraction condition (Fig. 1a, Supplementary Movie 2). We manipulated the opacity of irrelevant objects in the resource pool, thus increasing visual competition while searching for task-relevant objects. This allowed us to directly investigate the influences of visual distraction on several cognitive subcomponents of naturally unfolding behaviour. Working memory usage is generally low during natural behaviour[41–45] but increases when additional movement effort is required to access information in the external environment[42]. To experimentally induce more variability in participant's WM usage and investigate how the impact of distraction varies depending on locomotor demands, we therefore also included two movement effort conditions (Fig 1a, Supplementary Movie 3).

To foreshadow, this study makes three distinct contributions. First, we replicate that reliance on WM increased as sampling information from the environment required increased locomotion[42]. Second, we develop analytical tools that segment continuous behaviour into its core subcomponents. Third, we illustrate how specific changes in sensory parameters of the environment can have multiple cascading consequences for processes that are contingent on the affected subcomponent.

## Methods
### Participants
Thirty participants (Mean age = 24.5, range = 18–32, 22 women, 8 men, 0 other, 26 right-handed, 4 left-handed, all self-reported) were recruited at Goethe University Frankfurt ($n = 20$) and the University of Oxford ($n = 10$). All participants had a normal or corrected-to-normal vision (contact lenses; including colour vision) and provided informed consent prior to participating in the study. Participants recruited at Goethe University were compensated with course credit. At the University of Oxford, participants received £10/h as compensation. Participants completed an identical protocol at two testing days ($T_1$- $T_2$: 7–12 days). Participants were only included if they completed both testing days ($n = 1$ participant was replaced following non-participation on the second testing day) and data from both timepoints were combined for all reported analyses. The research protocol was approved by the local ethics committee of the Faculty of Psychology and Sport Sciences at Goethe University Frankfurt as well as the Central University Research Ethics Committee, University of Oxford (#R64089/RE001). Participants gave informed consent, and all relevant ethical regulations were followed throughout the study protocol.

Given that most dependent measures within our VR approach are unique to the present study, effect sizes to run a-priori power analyses were difficult to determine. Therefore, sample size planning was guided by a previous study using a similar VR paradigm[42], combined with a high trial-number approach to maximise the number of observations per participant. We ran simulation-based power analyses for the outcome variables *attributes used in memory*, *display completion time*, and *model viewing time* based on data from Draschkow and colleagues[42]. We explored power over a range of simulated effect sizes for the main effect of distraction (for details see shared materials). Thirty participants, each completing 224 trials, yielded power > 90% to detect an absolute mean difference of 0.04 attributes used in memory, 0.7 seconds in display completion time, and 8 milliseconds in model viewing time between distraction conditions. We did not pre-register the study protocol.

### Apparatus and virtual environment
Participants were equipped with an HTC Vive Tobii Pro VR integration with a built-in binocular eye tracker and one wireless HTC Vive Controller in their dominant hand. The head-mounted display (HMD) consisted of two 1080 ×1200 pixel resolution OLED screens (refresh rate = 90 Hz, field-of-view = 100°horizontally 3 ×110° vertically). Gaze was tracked at a sampling rate of 90 Hz (refresh rate of HMD) and an accuracy of approximately 0.5° visual angle. Gaze position in 3D space was obtained by intersecting the gaze vector with objects in the VR environment. Identical equipment was used at both locations (Frankfurt and Oxford).

We tracked the location in space for both the HMD and hand-held controller using two Lighthouse base stations emitting infrared pulses (60 per second) detected by the sensors in the devices (37 infrared sensors in the HMD and 24 in the controller). Tracking was further optimized by an accelerometer and a gyroscope embedded in the HMD, resulting in sub-millimetre precision.

A trigger button (operated with the index finger) and grip button (operated with the thumb) on the wireless controller were used for interacting with the environment. Participants pulled, held, and released the trigger button to grab, move, and place objects. The grip buttons were used to advance to the next stage of the experiment (e.g., start a new trial, and start calibration of the eye-tracker).

The virtual environment consisted of two alternating 400 × 400-cm rooms with a ceiling height of 240 cm: An Instruction room and a Trial room (see Supplementary Movie 2 and 3). The Instruction room displayed instructions and a fixation cross on the front-facing wall. A blue square on

the ground indicated the centre of the room. Participants could only advance to the next Trial room if positioned on the blue square and when the gaze intersected with the fixation cross. The Trial room contained three task-relevant stations: A model display, a resource pool, and a workspace. The model display consisted of 18 square white placeholders (arranged in a grid of 3 rows of 6 placeholders). Eight placeholders were occupied by pictures of the target objects in a random configuration. Above the model, a timer counting backwards from 45 seconds was displayed. Within the resource pool, participants would find an arrangement of 24 cubes (10 × 10 cm), organized in three rows of 8 cubes, as viewed from a frontal perspective, including the 8 targets seen in the model and an additional 16 distractors. The cubes were overlaid with images of objects from the Novel Object and Unusual Name Database[46], which had the advantage of being naturalistic while also unfamiliar and difficult to verbalize. Images for both target and distractor objects were selected from a stimulus pool of 60 images, for each display and participant. The location and identity of target objects in the model display was pseudo-randomized, ensuring that a specific arrangement would not repeat across trials for any given participant. The workspace consisted of an empty grid of 18 white placeholders mirroring the configuration of the model display.

The virtual environment and experiment were programmed and run in Unity (version 2019.3; Unity Technologies) using the SteamVR Unity plugin (version 1.2.10; Valve Corporation) on a computer operated with Windows 10.

## Procedure and task

Upon arrival, participants gave informed consent, were screened for vision requirements, and then familiarized with the HMD and the wireless controller.

During the task, participants had to copy the arrangement of objects seen in a model display by picking up the corresponding objects from the resource pool and placing them at the correct corresponding location in the workspace (see Fig. 1b and Supplementary Movie 1). Following a correct placement, the placed object "locked into" the workspace, and green contours indicated correct placement. Replacement objects (i.e., identical to the placed object) reappeared in the resource pool following correct placement to keep the number of objects in the resource pool constant. In case of an erroneous placement, the placeholder would light up red and the object would not lock into the workspace, nudging participants to correct their mistake. Participants could not pick up another object until the previously picked-up object was placed correctly: all other objects in the resource pool disappeared until the not-yet (correctly) placed object was placed correctly or brought back to the resource pool. This ensured that a) participants corrected their mistake before continuing with the task, preventing any flow-over of mistakes into the next task action and b) completed the task in a sequential object-by-object manner, prohibiting them from moving multiple target objects from the resource pool to the workspace before placing them in their corresponding placeholders.

No further instructions were provided apart from solving the task as accurately as possible, allowing participants to structure their behaviour freely. Trials timed out after 45 seconds, and participants were informed about their remaining time through a backwards counting timer above the model.

Critically, we manipulated the task environment in two ways. First, we varied the amount of visual distraction while searching for task-relevant objects in order to investigate the effect of the visual distraction on different subcomponents of behaviour. Specifically, we changed the opacity of distractor objects in the resource pool (Fig. 1 and Supplementary Movie 2), manipulating the discriminability between targets and distractors, and therefore distractibility, while participants searched for and picked up target objects. In the low-distraction condition, distractor objects were overlaid with a white plane set to 20% transparency. In the high-distraction condition, distractor objects appeared with the same opacity as target objects. We will henceforth denote this manipulation as distraction. Second, the model's location was varied (0° or 90° relative to the workspace, Fig. 1a and

Supplementary Movie 3), thus manipulating the movement effort associated with encoding information from the model[42]. Prior work demonstrated that reliance on memory during naturalistic tasks is generally low[42,43] but increases when accessing information in the task environment becomes more effortful[42–45,47]. For example, as the distance to the model increases (i.e., requiring greater movement effort to look back to the model), participants spend more time encoding from the model as well as relying on WM more[42]. Replicating previous work[42], we manipulated movement effort to experimentally induce more variability in how much participants relied on WM, allowing us to observe the effects of visual distraction during a broader range of naturalistic memory usage.

On both testing days, participants completed a minimum of four practice trials to familiarise themselves with the task and the VR environment. The practice phase ended if participants were able to finish the last practice trial within the time constraint of 45 seconds. Otherwise, one more practice trial was added until this condition was met. Afterwards, participants completed 8 blocks of 14 displays. Each block consisted of trials from a single condition (e.g., 90° and high distraction) and was further divided into two sub-blocks (7 displays each) differing in the relative positions of the model and resource displays and, therefore, direction of required movement. That is, participants had to either turn to the right or left to find the resource pool. The order of sub-blocks was randomized within each block and participant. A 5-point calibration was performed after each block or manually whenever participants failed the fixation-check protocol.

The 8 blocks were further divided into two sessions consisting of 4 blocks each. A mandatory break (5–10 minutes) was included between sessions during which participants took off the HMD and could rest. Both sessions contained all combinations of conditions, and the order of blocks was randomized within each session. In total, participants completed 112 displays per testing day (28 displays and 224 copied objects per condition, 224 displays overall). The experiment lasted approximately 90–120 minutes on each testing day.

## Data processing and measures

Frame-by-frame (90 Hz) data were written into csv-files during recording. For each frame, we recorded which objects in the VR environment were looked at, the location in space of both HMD and controller, as well as all relevant interactions with the environment (e.g., if and which object was grabbed). For the purpose of data analysis, we then segmented and summarised our data in order to extract metrics for the subcomponents highlighted in Fig. 1b.

**Overall behaviour.** We quantified overall behaviour through *display completion time* and *total head movement*. Display completion time was calculated by summing up all frame durations from trial start to end. The total head movement per display was determined by first calculating the Euclidian distance of the coordinates of the HMD between subsequent frames, which were then all summed up from trial start to end.

**Encoding.** To quantify encoding, we first identified the periods in which participants looked at the model (i.e., the gaze intersected with the model). The start of an encoding period was set as the frame when gaze first intersected with the model. An encoding period was considered finished when the gaze stopped intersecting with the model and gaze did not return to the model within 25 frames (~250 ms). To further qualify as an encoding period, participants had to have either looked at the resource pool or workspace before their gaze returned to the model. After determining periods of encoding, *model viewing time* was calculated by summing up frame durations from encoding start to end. The *number of model viewings* per display was determined by counting periods of encoding within each trial (i.e., display). *Total encoding per display* was calculated by summing up all model viewing times within each trial. We additionally determined the *number of targets encoded* by counting the number of individual target slots that gaze intersected with for at least two consecutive frames (~20 ms) within each period of encoding.

**Visual search**. Similarly, we quantified visual search by first identifying periods in which participants searched for objects in the resource pool. The start of a visual search period was set to the frame when the gaze first intersected with the resource pool. We considered a visual search period to be finished when participants picked up an object from the resource pool. We then calculated *search time* by summing up frame durations from search start to end. The *number of looked-at objects* during each search was determined by counting the number of objects (targets and distractors separately) in the resource pool the gaze intersected with for at least two consecutive frames (approx. 20 ms) within each search. We additionally determined *object viewing time*, which was calculated by determining periods within one search where the gaze intersected with a specific object in the resource pool (for at least two consecutive frames) and summing up the frame durations while the gaze intersected with this specific object.

**WM usage**. WM usage was calculated by segmenting task behaviour into sequences, which each started and ended with an encoding period. For example, a sequence could consist of an initial gaze on the model, followed by a gaze to the resource pool and workspace before concluding with another gaze on the model, which would mark the end of the current and the beginning of the next sequence. To further qualify as a sequence, participants had to have either looked at the resource pool or workspace in between gaze on the model. Since the successful copying of one object minimally requires the usage of two representational attributes (the object's identity for finding and picking it up, and its location for correctly placing it), *attributes used in WM* were calculated by counting the correct pick-ups and placements of target objects within each sequence.

**Sensorimnemonic decisions**. We identified two discrete sensorimnemonic decisions (i.e., decisions of whether to rely on memory versus information in the external world) within the object copying task. First, after picking-up a target object from the resource pool, participants had to decide between relying on the (previously encoded) location of the object or gathering information from the model to guide their next placement (i.e., a *location-related* decision). Consequently, location-related decisions were locked to the frame (90 Hz) after a target had been picked-up. We then determined if participants re-fixated the model before placing the picked-up object. If participants placed the picked-up object without re-fixating the model first, participants decided to rely on memory to place the object. Second, after placing a target object in the workspace, participants had to decide between using memory to pick up the next object or first to re-fixate the model to encode identity information (i.e., *identity-related* decision). Identity-related decisions were therefore locked to the frame after an object placement. We then identified if participant re-fixated the model before picking-up the next target. If participants did not re-fixate the model, participants decided to rely on memory to pick-up the object. Sensorimnemonic decisions were further indexed by their position within a WM usage sequence. That is, if participants encoded, then searched and picked-up an object, the following location-related decision would be the first sensorimnemonic decision within this behavioural sequence. If participants decided to rely on memory to place the picked-up object, the following identity-related decision would be the second sensorimnemonic decision within this behavioural sequence. The *probability of using memory* for each sensorimnemonic decision (i.e., for each position within the behavioural sequence separately) was then calculated as the proportion of sensorimnemonic decisions during which participants decided to rely on memory divided by the total number of decisions.

**Errors**. We identified two types of errors: Identity and location errors. Any pick-up of a non-target object (i.e., distractor object or previous target that has already been placed) from the resource pool was counted as an identity error. For location errors, we considered all instances in which participants attempted to place a picked-up *target* object (i.e.,

attempted placements of non-targets were not considered). All placements in an incorrect workspace slot were counted as a location error. Errors were further indexed by their position within their behavioural sequence as well (i.e., for details see WM usage and sensorimnemonic decisions above). The *percentages of location/identity errors* at each position within the behavioural sequences were then calculated as the proportion of incorrect placements/pick-ups of the total number of placements/pick-ups.

### Data analysis

All data were pre-processed and analysed in the R statistical programming language (version 4.2.2;[48]) using RStudio (version 2023.6.2.561;[49]). The lme4 package[50] was used to run linear mixed models (LMMs) and generalised linear mixed models (GLMMs), which were all fitted with the restricted maximum likelihood criterion.

To meet LMM assumptions, distribution and power coefficient of all continuous dependent variables were inspected using the MASS package[51] and the Box-Cox procedure[52]. As a result, all continuous dependent variables except total encoding time were log-transformed. Sum contrasts were defined for movement effort conditions (0° vs. 90°) as well as the distraction conditions (high vs. low). Accordingly, the grand mean of the dependent measure serves as the intercept, and slope coefficients can be interpreted as main effects. Additional continuous predictor variables were z-transformed (scaled and centred).

Model selection always started with the maximal random-effects structure[53]. If not specified otherwise, this included subject random intercepts as well as by-subject random slopes for movement effort and distractibility as well as their interaction. We then identified potential overparameterization in each model by using a principal component analysis (PCA) of the random effect variance-covariance estimates and removed random slopes that were not supported by the PCA and did not contribute significantly to the goodness of fit in a likelihood ratio (LR) test[50].

For LMMs, we report regression coefficients β with the t statistic. P-values are calculated with Satterthwaite's degrees of freedom method using the lmerTest package[54]. For GLMMs, we report β with the z statistic and p-values are based on asymptotic Wald tests. For all models, we apply a two-tailed criterion corresponding to a 5% error criterion for significance. Significant interactions were broken down by examining planned pairwise comparisons using the emmeans package[55] with default Tukey-adjusted p-values, as well as through conditional effects using the ggeffects package[56]. Where applicable, differences in means between planned comparisons were compared using paired t-tests[48]. All pairwise comparisons were additionally followed up by paired Bayesian t-tests using the BayesFactor package[57], with default priors ($r = 707$) to test the null hypothesis ($m = 0$) against an alternative hypothesis suggesting a non-zero effect size ($r = 0.707$). In line with reporting policies, we additionally report $\eta_p^2$ for LMMs and the corresponding 95% confidence intervals using the effectsize package[58]. Note that for GLMMs, β acts as a standardised effect size measure and we report Wald's confidence intervals referring to the reported β. For all pairwise comparisons, we report Cohen's d and the corresponding 95% confidence intervals as an effect size measure[58]. Visualizations were done using the ggplot2 package[59] and standard errors for plots were computed using the Rmisc package[60].

**Overall behaviour**. Only trials that were completed before the timeout (i.e., < 45 seconds) were included in the analyses of overall behaviour (101 trials excluded across all participants; 1.5% of data). The effect of movement effort, distraction, and their interaction on display completion time and total head movement was analysed using LMMs and we report outcomes of the full models. For a complete report of the results, see Supplementary Tables 2 and 3.

**Encoding**. For all analyses concerning encoding, we excluded encoding periods that were not followed by either gaze on the workspace or resource pool (2.6% of encoding periods). Since the model appeared in

front of the participant at trial start and participants had to first orient themselves in the trial room, we further excluded the first encoding period in each display (additional 9.4% of encoding periods). Additionally, encoding periods with a model viewing time that lasted less than 50 ms (0.5% excluded) were excluded. Next, we removed encoding episodes with a model viewing time of more than 3.5 standard deviations above the individual mean across all conditions (additional 1.4%), leaving 60,976 encoding periods that entered analysis. For total encoding time and model viewings per trial, only trials that have been completed before timeout were included in the analysis (see overall behaviour above). The effect of movement effort, distraction, and their interaction on total encoding time and model viewing time was then analysed using LMMs. Model viewings per trial and the number of encoded objects were analysed using GLMMs with a Poisson distribution. For total encoding time, model viewing time, and model viewings per display, we report outcomes of the full models. The random-effect structure for the best fitting model for number of targets encoded included a subject intercept and by-subject random slope for movement effort. For a complete report of the results, see Supplementary Tables 4 and 5.

**Visual search**. Search periods with a search time of less than 50 ms (<0.1% excluded) and search periods that ended with a pick-up error (additional 2.0% of searches) were excluded from all analyses on the visual search subcomponent. Additionally, we removed search periods with a search time of more than 3.5 standard deviations above the individual mean across all conditions (additional 1.2%), leaving 55,093 searches that entered analysis. The effects of movement effort, distraction, and their interaction on search time and object distractor viewing times (targets and distractors separately) were then analysed using LMMs. The number of looked-at objects (targets and distractors separately) was analysed using GLMMs with a Poisson distribution. For all analyses, we report outcomes of the full models. For a complete report of the results, see Supplementary Tables 6 and 7.

**WM usage**. Given the tight link between encoding periods and the behavioural sequences used to quantify WM usage, we included the same data that entered analyses for encoding in our analyses for WM usage (see exclusion criteria above). The effect of movement effort, distraction, and their interaction on the number of attributes used in WM was then analysed using GLMMs with a Poisson distribution. The random-effect structure for the best-fitting model included a subject intercept and by-subject random slopes for movement effort and distraction. Additionally, we computed the probability of different numbers of attributes used in WM. Differences in means between distraction conditions were analysed nested in movement effort conditions, using planned paired pairwise $t$-tests as well as paired Bayesian $t$-tests. We further investigated the relationship between the number of attributes used in WM and the preceding encoding episode. Here, model viewing time entered the fixed effect structure as an additive predictor. The random-effect structure for the best-fitting model with this added predictor again included a subject intercept and by-subject random slopes for movement effort and distraction. For a complete report of the results, see Supplementary Tables 8–10.

**Sensorimnemonic decisions**. Given the unconstrained nature of the VR task, sensorimnemonic decisions could be preceded by vastly different behaviour. Nonetheless, to compare the different types of sensorimnemonic decisions across distractor conditions, we aimed to align the decisions within their behavioural sequences as stringently as possible. To do so, we focused on decisions within sequences during which participants searched and picked up an object directly after encoding from the model (70% of sequences in 90° movement effort condition). That is, the first sensorimnemonic decision in this type of behavioural sequence was a location-related decision (i.e., decide to place the picked-

up object vs. encode location from the model first; see Supplementary Notes 4 for analysis on decisions within the remaining 30% of sequences that started with placement and an identity-related decision). Further, decisions were only included if the initial encoding period of the decision sequence was within the inclusion criteria for encoding periods (see encoding above). Decisions were also excluded if we could not identify and match the search period preceding the first location-related decision or if this search period was outside of the exclusion criteria for search (see visual search above). Additionally, decisions were excluded if participants were disrupted during the behavioural sequence leading up to the decision (e.g., by making an error that had to be corrected), if participants decided to rely on memory but made a mistake (e.g., decided to place a picked-up object but placed the object at the wrong location), or if the decision was not followed by either encoding from the model or placement/pick-up (e.g., the trial ended or following action could not be identified due to idiosyncratic behaviour). Since participants predominantly relied on only one attribute when movement effort was low (i.e., participants only rarely reached the second identity-related decision), we focused on decisions in trials with high movement effort. Here, participants relied on up to four attributes often enough for us to consider three consecutive sensorimnemonic decisions. Overall, 13,198 decisions were included in the analysis for the first location-related decision, 7833 decisions were included in analysis regarding the second identity-related decisions, and 1670 decisions remained for the third location-related decision. The effect of distraction on the probability of using memory (i.e., sampling from model vs. direct placement or pick-up using memory) was then investigated using GLMMs with a Binomial distribution. For the first location-related decision and the second identity-related decision, we report outcomes from the full model. That is, the random-effect structure included a subject intercept and by-subject random slopes distraction. The random-effect structure for the best-fitting model for the third location-related decision only included a subject intercept. For a complete report of the results, see Supplementary Table 11.

**Errors**. Analyses of error rates were based on the same data used to analyse sensorimnemonic decisions. That is, we again focused on errors within sequences during which participants searched and picked up an object directly after encoding from the model display (see Supplementary Notes 4 for analysis of errors within sequences that started with a placement). In comparison to analyses on sensorimnemonic decisions, however, sequences with incorrect placements and pick-ups were not excluded. Given that incorrectly picked-up objects inadvertently lead to a placement error if participants attempted to place a non-target object (combined with the general disruption caused by the need to correct errors), sequences were excluded after the first error occurred. Any subsequent errors within the same behavioural sequence therefore did not enter data analysis to keep location and identity errors pure. Additionally, since errors had to be corrected immediately and therefore differed from correct placements/pick-ups in how participants could proceed with the task, placement, and pick-ups were included irrespective of the following action. This left us with 13,825 pick-ups immediately following encoding from the model (i.e., first pick-up), 10,080 first placements, 1845 second pick-ups, and 1523 second placements. The effect of distraction on location or identity errors (i.e., correct vs. incorrect placement or pick-up) was then investigated using GLMMs with a Binomial distribution. Separate models were run for the first and second pick-up or placement in the behavioural sequences respectively. The random-effect structure for all best fitting models only included a subject intercept. For a complete report of the results, see Supplementary Table 12.

## Reporting summary

Further information on research design is available in the Nature Portfolio Reporting Summary linked to this article.

## Results

In this section, we first consider the consequences of visual distraction on summary measures of overall temporally extended behaviour. We then dissect the extended behaviour into standalone cognitive subcomponents and separately focus on the effects of distraction on encoding, visual search, WM usage, and sensorimnemonic decision-making. In the Results section, we highlight the most relevant findings. The full outputs for the mixed-effects modelling approach can be found in the Supplementary Tables 1–12.

### Distraction slows down behaviour and increases costly body movements

To test whether encountering more visual distraction while searching for target objects impedes overall behaviour, we calculated the time required to copy each display (i.e., *display completion time*, Fig. 2a, top) as well as the total head movement associated with copying a display (i.e., *total head movement*, Fig. 2b).

Both distraction and movement effort produced changes in overall behaviour. High distraction significantly slowed display completion time ($\beta = 0.05$, $SE = 0.003$, $t = 18.23$, $p < 0.001$, $\eta_p^2 = 0.92$, $CI_{95\%} = [0.86, 0.95]$) and led to more head movement ($\beta = 0.02$, $SE = 0.002$, $t = 9.15$, $p < 0.001$, $\eta_p^2 = 0.75$, $CI_{95\%} = [0.56, 0.84]$), compared to the low distraction condition.

Unsurprisingly, high movement effort also led to more head movement while copying displays ($\beta = -0.06$, $SE = 0.003$, $t = -22.07$, $p < 0.001$, $\eta_p^2 = 0.94$, $CI_{95\%} = [0.91, 0.96]$). Additionally, high movement effort also led to slower display completion times ($\beta = -0.08$, $SE = 0.003$, $t = -24.79$, $p < 0.001$, $\eta_p^2 = 0.95$, $CI_{95\%} = [0.92, 0.97]$), replicating previous findings[42]. For display completion time, movement effort and distraction interacted ($\beta = 0.01$, $SE = 0.002$, $t = 4.00$, $p < 0.001$, $\eta_p^2 = 0.36$, $CI_{95\%} = [0.10, 0.57]$): high distraction slowed down global behaviour more strongly when movement effort was low ($t = 2.05$, $df = 29$, $p = 0.02$; $d = 0.37$, $CI_{95\%} = [0, 0.74]$; $BF_{10} = 1.21$), but the difference in means (0.52 s) was small relative to the overall display completion times and the follow-up Bayesian t-test provided no credible evidence in favour of the alternative hypothesis.

### Distraction influences overall encoding demands

As a proxy for encoding, we next isolated the periods participants spent viewing the model during the selection of to-be-copied objects (Fig. 1b). Both distraction and movement effort affected encoding demands during the task, although in distinct ways.

High distraction was associated with longer total encoding per display ($\beta = 0.14$, $SE = 0.03$, $t = 5.04$, $p < 0.001$, $\eta_p^2 = 0.47$, $CI_{95\%} = [0.20, 0.65]$; Fig. 3a). Specifically, participants encoded from the model *more often* when distraction was high ($\beta = 0.03$, $SE = 0.01$, $z = 5.19$, $p < 0.001$, $CI_{95\%} = [0.02, 0.04]$; Fig. 3c), but we observed no meaningful difference in the time participants spent encoding during each encoding episode ($\beta = 0.01$, $SE = 0.003$, $t = 1.90$, $p = 0.07$, $\eta_p^2 = 0.11$, $CI_{95\%} = [0, 0.34]$; Fig. 3b) or in the number of objects encoded from the model during each episode (see Supplementary Notes 1 and Supplementary Fig. 1). The effect of distraction on

model viewing time was subtly moderated by movement effort ($\beta = 0.01$, $SE = 0.003$, $t = 2.34$, $p = 0.03$, $\eta_p^2 = 0.17$, $CI_{95\%} = [0.01, 0.41]$). Planned comparisons revealed that model viewing time between distractor conditions did not differ when movement effort was high ($\beta < 0.0001$, $SE = 0.01$, $z = 0.01$, $p = 1$, $d = -0.02$, $CI_{95\%} = [-0.38, 0.34]$), with a follow-up Bayesian t-test ($BF_{10} = 0.20$) providing moderate evidence in favour of the null hypothesis. During the low movement effort condition, participants encoded slightly longer when distraction was high compared to low ($\beta = 0.02$, $SE = 0.01$, $z = 2.86$, $p = 0.02$, $d = -0.52$, $CI_{95\%} = [-0.83, -0.13]$), with a follow-up Bayesian t-test ($BF_{10} = 5.19$) providing moderate evidence in favour of the alternative hypothesis. However, the difference between predicted means during low movement effort was very small (~9.5 ms) relative to the overall model viewing time.

In comparison, movement effort influenced overall encoding demands by affecting both encoding frequency and durations. Interestingly, participants devoted less time in total encoding per display ($\beta = 0.28$, $SE = 0.05$, $t = 5.20$, $p < 0.001$, $\eta_p^2 = 0.48$, $CI_{95\%} = [0.21, 0.66]$; Fig. 3a) when movement effort was high. This is the result of encoding longer during each encoding episode ($\beta = -0.19$, $SE = 0.02$, $t = -11.44$, $p < 0.001$, $\eta_p^2 = 0.82$, $CI_{95\%} = [0.68, 0.89]$; Fig. 3b) in combination with encoding from the model less often ($\beta = 0.25$, $SE = 0.02$, $z = 15.37$, $p < 0.001$, $CI_{95\%} = [0.22, 0.29]$; Fig. 3c). This replicates previous findings showing that participants "loaded-up" more information in one go when the task was more effortful (see Supplementary Notes 1)[42].

Given the temporally-extended nature of each trial, we conducted a post-hoc analysis into potential temporal effects in the encoding subcomponent, showing that the effects of distraction and movement effort go above and beyond any temporal task effects (see Supplementary Notes 2).

### Distraction interferes with visual search

To investigate the effect of distraction on the visual search subcomponent (Fig. 1b), we isolated periods during which participants searched for relevant objects in the resource pool (Fig. 4). Encountering more distracting distractor objects while searching for target objects strongly interfered with visual search.

Participants were slower to find objects when faced with high distraction ($\beta = 0.1$, $SE = 0.004$, $t = 26.16$, $p < 0.001$, $\eta_p^2 = 0.96$, $CI_{95\%} = [0.93, 0.97]$; Fig. 4a). Specifically, participants looked at more distractors ($\beta = 0.29$, $SE = 0.01$, $z = 33.76$, $p < 0.001$, $CI_{95\%} = [0.27, 0.31]$; Fig. 4b) and viewed distractors longer ($\beta = 0.09$, $SE = 0.003$, $t = 31.12$, $p < 0.001$, $\eta_p^2 = 0.97$, $CI_{95\%} = [0.95, 0.98]$; Fig. 4c) when distraction was high. In high distraction trials, participants also looked at target objects longer ($\beta = -0.02$, $SE = 0.01$, $z = -2.96$, $p = 0.006$, $\eta_p^2 = 0.23$, $CI_{95\%} = [0.02, 0.47]$), but they looked at subtly more target objects during low compared to high distraction trials ($\beta = -0.02$, $SE = 0.01$, $z = -2.96$, $p = 0.003$, $CI_{95\%} = [-0.04, -0.0]$).

Movement effort had no significant impact on search time ($\beta = 0.0003$, $SE = 0.004$, $t = 0.09$, $p = 0.93$, $\eta_p^2 < 0.001$, $CI_{95\%} = [0, 0.07]$), but it did interact with distraction ($\beta = 0.006$, $SE = 0.003$, $t = 2.32$, $p = 0.03$, $\eta_p^2 = 0.16$,

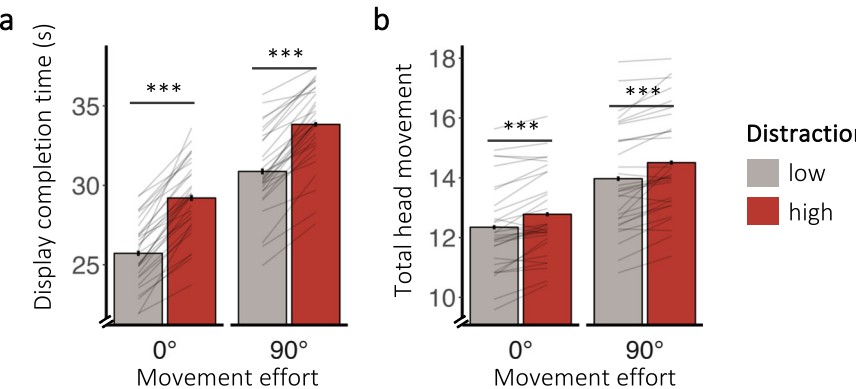

**Fig. 2 | Effects of distraction on global behaviour.** **a** Influence of distraction on display completion time and **b** total head movement, which served as a proxy for global behavioural performance. Error bars depict the standard error of the mean ($N = 30$). Lines show individual participant data. The symbols *, **, and *** in the figure denote statistical significance with p-values less than 0.05, 0.01, and 0.001, respectively.

CI$_{95\%}$ = [0.01, 0.4]). However, planned pairwise comparisons revealed no statistically significant differences in search times when distraction was low (β = –0.01, SE = 0.01, z = –1.22, p = 0.62, d = –0.22, CI$_{95\%}$ = [–0.58, 0.14]; BF$_{10}$ = 0.38) or when it was high (β = 0.01, SE = 0.01, z = 1.44, p = 0.47, d = 0.26, CI$_{95\%}$ = [–0.11, 0.62]; BF$_{10}$ = 0.48). Conversely, Bayes Factors yield no credible evidence for the absence of such a difference in either condition. We further found no statistically significant differences between movement effort conditions in the number of distractors (β = 0.01, SE = 0.01, z = 0.66, p = 0.51, CI$_{95\%}$ = [–0.01,0.02]) or targets (β = –0.001, SE = 0.01, z = –0.18, p = 0.85, CI$_{95\%}$ = [–0.01, 0.01]) participants looked at during search (see Supplementary Tables 5 and 6 for analysis on target and distractor viewing times). A post-hoc control analysis showed that the effects of distraction and movement effort go above and beyond any temporal task effects (see Supplementary Notes 2).

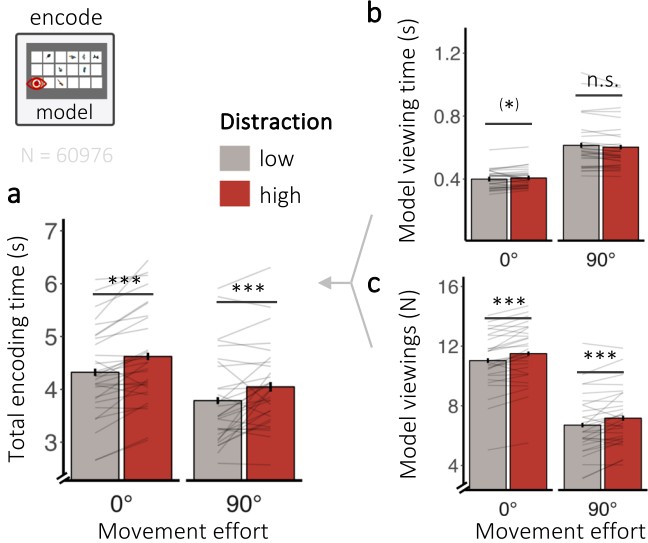

**Fig. 3 | Distraction affects overall encoding demands. a** Influence of distraction on total encoding time (i.e., time spend encoding while completing one display, left), **b** model viewing time (i.e., time spent encoding from the model each time it is viewed), and **c** the number model viewings while copying one display. The data are depicted as a function of the movement effort condition. Error bars depict the standard error of the mean (N = 30). Lines show individual participant data. N shows the number of encoding episodes included in the analysis. The symbols *, **, and *** in the figure denote statistical significance with p-values less than 0.05, 0.01, and 0.001, respectively.

### Distraction decreases memory usage

Returning to the baking example: Would we rely on our memory less when distraction is high? Compared to traditional lab studies, in our task participants could choose when to rely on memory representations or when to sample information from the external environment (Fig. 1b). We capitalised on our recent discovery that the coordination between encoding information from the external world and relying on information from memory during natural behaviour can be captured by a metric we refer to as WM usage[42]. Deriving an implicit measure for WM usage, therefore, allowed us to capture a fundamental co-ordinational aspect of complex behaviour.

In our task, successful copying of one object minimally requires the usage of two representational attributes: the object's identity for finding and picking it up, and its location for correctly placing it (Figs. 1b and 5a). Here, *picking up* a target object indicates that one attribute (i.e., object identity) has been used in WM. If a participant then *places* the object in the workspace without looking back to the model, a second attribute (i.e., object location) has been used to guide behaviour.

As intended, the average number of attributes used in WM increased with movement effort (β = –0.21, SE = 0.01, z = –18.17, p < 0.001, CI$_{95\%}$ = [–0.23, –0.18])[42,44], allowing us to observe a broader range of natural WM usage (Fig. 5b). Further replicating findings from Draschkow et al.[42], participants mostly relied on only one attribute in WM when locomotive effort was low but were more likely to rely on 2 attributes in WM when movement effort was high (Fig. 5c). This shift towards more WM usage was supported by participants encoding longer each time they viewed the model, with encoding time positively predicting subsequent WM usage (see Supplementary Notes 1 and Supplementary Fig. 1), highlighting again the close relationship between encoding information from the external world and relying on information from memory.

Distraction also critically influenced the coordination between gathering information from the environment and using information in WM, although to a lesser extent. Encountering high distraction led to participants relying on their memory less (β = –0.03 SE = 0.004, z = –7.02, p < 0.001, CI$_{95\%}$ = [–0.03, –0.01]). Here, distraction interacted with movement effort (β = 0.01, SE = 0.0043, z = 2.23, p = 0.03, CI$_{95\%}$ = [0.001, 0.01]): Distraction decreased WM usage more when movement effort was high (t = 3.05, df = 29, p = 0.002, d = 0.56, CI$_{95\%}$ = [0.17, 0.94]; BF$_{10}$ = 8.43). Comparing the probability of using different numbers of attributes in WM (Fig. 5c) further highlighted that the distraction-induced difference in WM usage was consistently driven by a small decrease in using 3 or 4 attributes (see Supplementary Table 10 for a full reporting of the pairwise comparisons).

Importantly, the flexibility in self-structuring behaviour during the task did additionally allow for potential strategy changes. That is, behavioural sequences could have a different structure, even if participants are using the same number of attributes in WM. For example, using two attributes in WM

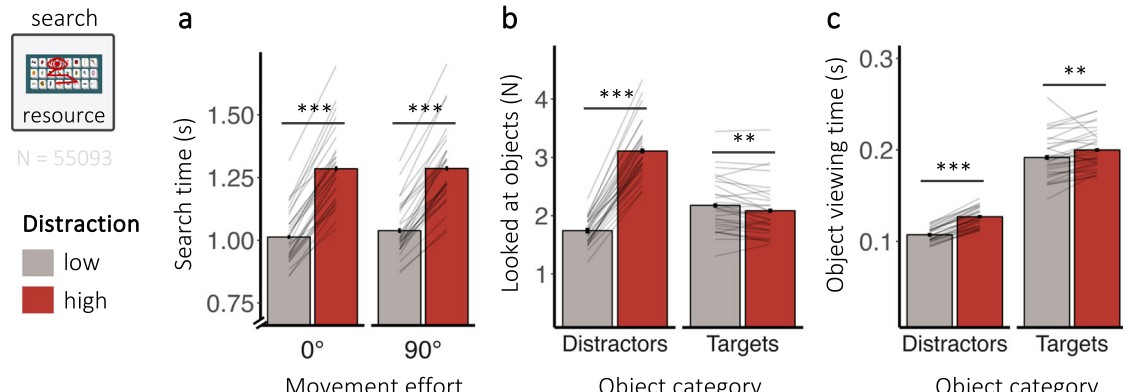

**Fig. 4 | Distraction interferes with visual search.** Influence of distraction on **a** search time for target objects, **b** the number of distractors and targets looked at during search (collapsed over movement effort condition, middle), and **c** the time spent looking at distractors or target objects during search (collapsed over movement effort condition, right). Error bars depict the standard error of the mean (N = 30). Lines show individual participant data. N shows the number of overall searches included in the analysis. The symbols *, **, and *** in the figure denote statistical significance with p-values less than 0.05, 0.01, and 0.001, respectively.

**Fig. 5 | Trade-off between reliance on WM and gathering information from the external world.**
**a** Our implicit metric for working memory (WM) usage: Copying each object requires its identity and location information (attribute) to be held in memory. Counting successful pick-ups (i.e., identity attribute used) and placements (i.e., location attribute used) in between model fixations provided a metric for the number of attributes used in WM.
**b** Average number of attributes used in WM in both movement effort conditions as a function of distraction, **c** distribution of attributes used in WM (%). Error bars depict the standard error of the mean (*N* = 30). Lines show individual participant data. N shows the number of overall sequences included in the analysis The symbols *, **, and *** in the figure denote statistical significance with *p*-values less than 0.05, 0.01, and 0.001, respectively.

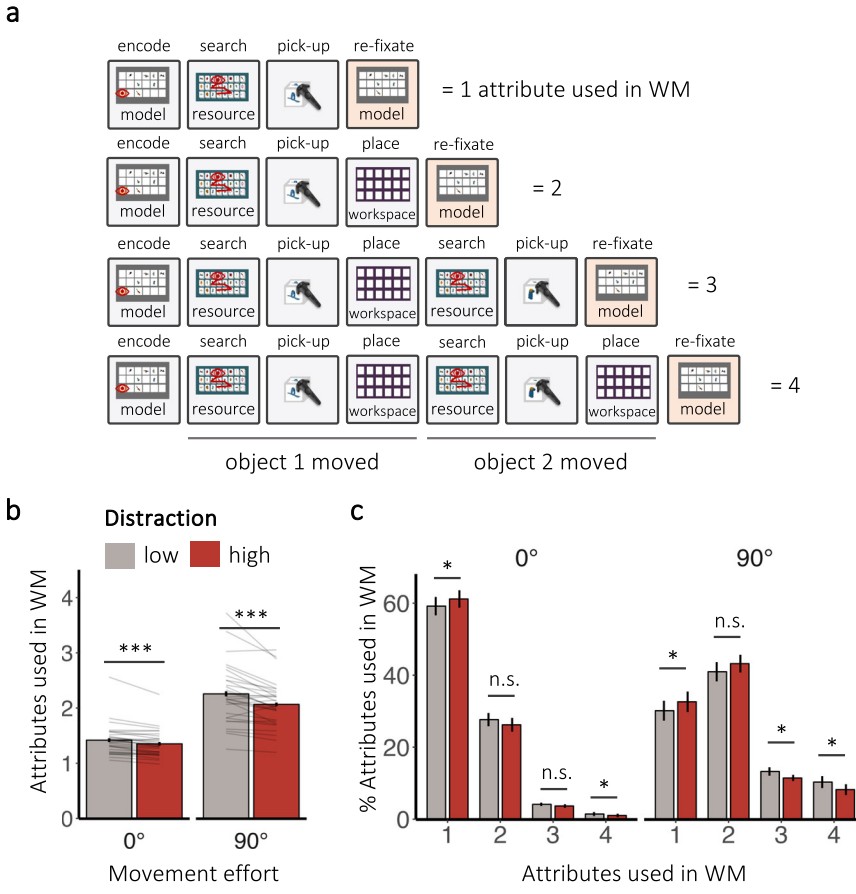

could be the result of first searching for and picking up a target object (i.e., identity feature used) and then directly placing this object (i.e., location attribute used). At the same time, participants could have used two attributes in WM by first placing an already picked-up object (i.e., participant encoded while holding an object in hand) before continuing to search for and pick-up another object. However, while visual distraction-affected WM usage, it did not lead to major strategy changes (see Supplementary Notes 3).

**Consequences of distraction on sensorimnemonic decisions**
Our findings from the encoding and WM usage metrics converge on a consistent pattern: When encountering high distraction, participants on average use a slightly lower number of attributes in WM. Instead, they encode from the external environment more frequently.

However, the coordination between encoding and WM usage ultimately results from distinct sensorimnemonic decisions. For instance, when participants pick up the first object, they must decide between relying on the previously encoded location of the object or gathering information from the model to guide their next placement action (i.e., a *location-related* decision, Fig. 1). Similarly, after placing an object, individuals must again decide between using memory to pick up the next object (i.e., *identity-related* decision) or first to re-fixate the model.

Natural behaviour is punctuated by instances of such sensorimnemonic decisions, but how they are impacted by distraction has not been studied before. Our VR approach enabled us to isolate these decision points during continuous behaviour, allowing us to trace the consequences of distraction all the way to the decision-making processes that gate memory usage. To increase sensitivity, we focused our analysis on trials with high movement effort, as participants predominantly relied on only one attribute when movement effort was low, precluding us from observing longer sequences of memory-guided behaviour (see overall WM usage depicted in Fig. 4b).

To anticipate, we identified two types of sensorimnemonic decisions, some of which were *robust* and others *affected* by our specific distraction manipulation (i.e., increased visual distraction from competing task-irrelevant object identities while searching for relevant objects).

Specifically, *identity-related* sensorimnemonic decisions – that is, decisions to pick up objects using initially encoded identity information versus re-encoding the object identity before picking them up – were impacted by high distraction. For these specific decisions, encountering high distraction led to participants deciding to rely on memory less. When faced with high distraction, participants were less likely to decide to pick up a second object (β = −0.23, *SE* = 0.05, *z* = −4.23, *p* < 0.001, $CI_{95\%}$ = [−0.33, −0.12]; $BF_{10}$ = 71.16; Fig. 6b), which would have required behaviour to be guided by using a second identity attribute from memory. Interference with identity-related sensorimnemonic decisions were further corroborated by additional analysis of other sequence types (see Supplementary Notes 4).

In contrast, we did not observe a similar effect of high distraction on *location*-related sensorimnemonic decisions – that is, decisions to place an object using its encoded location attribute versus re-encoding its location first. In these cases, encountering high distraction did not lead to participants deciding to rely less on memory. Distraction did not lead to statistically significant differences in the probability of relying on memory when placing the first object (β = −0.05, *SE* = 0.03, *z* = −1.39, *p* = 0.16, $CI_{95\%}$ = [−0.11, 0.02]; Fig. 6a), with a follow-up Bayesian *t*-test ($BF_{10}$ = 0.45) providing no credible evidence in favour of the null hypothesis. Further, the probability of placing the second object using its encoded location was not negatively affected by distraction. Participants were instead more likely to decide to rely on memory when encountering high distraction (β = 0.24, *SE* = 0.08, *t* = 3.12, *p* = 0.002, $CI_{95\%}$ = [0.09, 0.39]; Fig. 6c). However, a follow-up Bayesian *t*-test ($BF_{10}$ = 1.37) provided no credible evidence for the alternative hypothesis.

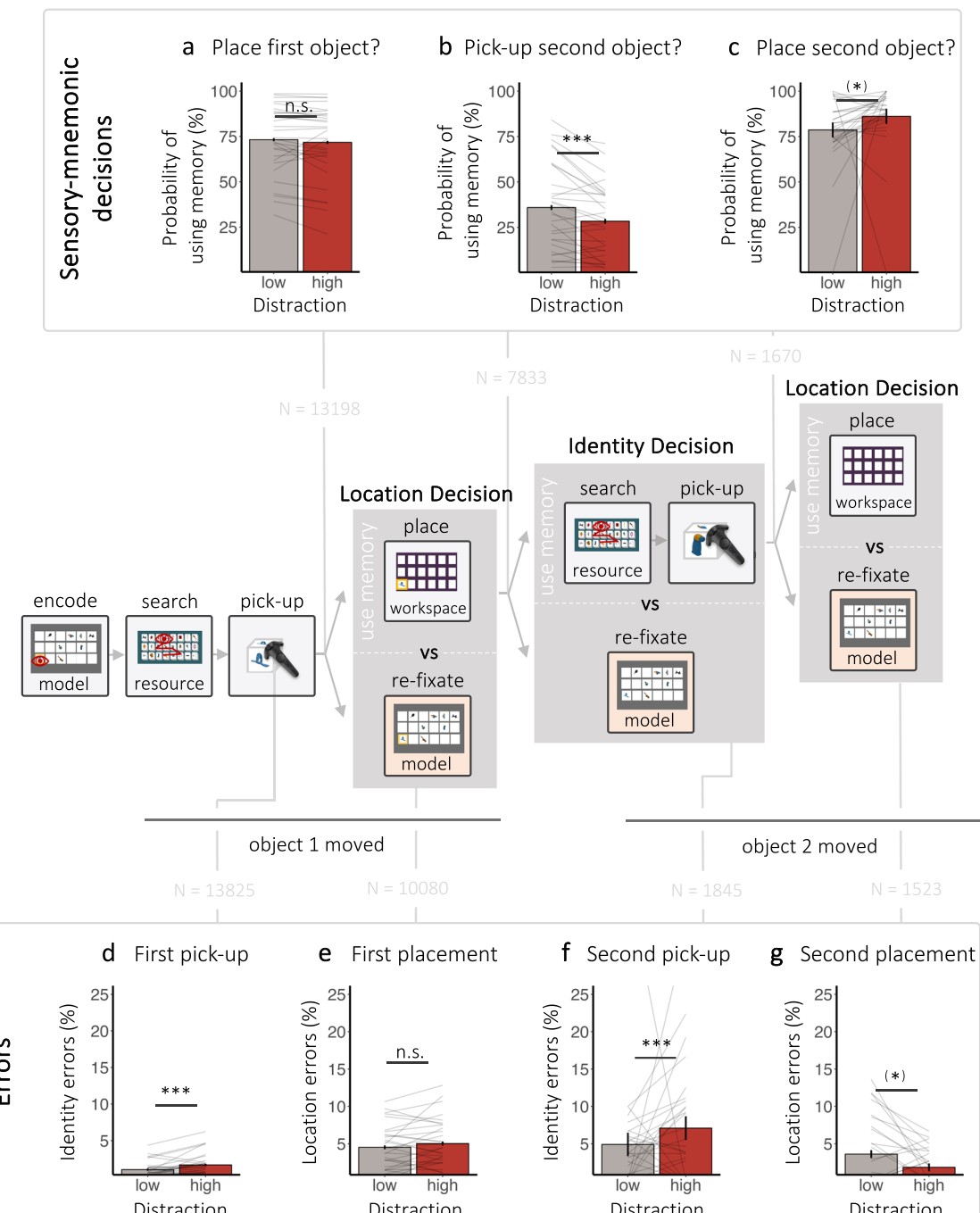

**Fig. 6 | Distraction influences sensorimotor decisions and errors.** The flow diagram depicts the sequence of sensorimnemonic decisions in our VR Object copying task. **a** The pick-up of the first object is followed by a location-related decision (i.e., Place object using memory or re-fixate model?). **b** If participants place the object using memory, placement is followed by an identity-related decision (i.e., Pick-up a second object or re-fixate model?). **c** If memory was used again, a second location-related decision would follow. Error bars depict standard error of the mean (a and b: $N = 30$, c: $N = 28$). Identity and location errors at different moments of behavioural sequences. Error bars depict standard error of the mean (**d** and **e**: $N = 30$, **f** and **g**: $N = 28$). All plots and analyses are based on the high movement effort condition. Lines show individual participant data. N shows the number of sensorimnemonic decisions or pick-ups/ placements included in the analyses. The symbols *, **, and *** in the figure denote statistical significance with $p$-values less than 0.05, 0.01, and 0.001, respectively.

Further, we observed how distraction in the form of competing object identity information interfered with memory guided behaviour through error rates (also see Supplementary Notes 4 and Supplementary Fig. 5). Participants made more identity errors (i.e., picked up non-target objects) when distraction was high, evident when picking up the first ($\beta = 0.27$, $SE = 0.08$, $z = 3.64$, $p < 0.001$, $CI_{95\%} = [0.13, 0.42]$; $BF_{10} = 5.63$; Fig. 6d) and second object ($\beta = 0.36$, $SE = 0.10$, $z = 3.51$, $p < 0.001$, $CI_{95\%} = [0.16, 0.56]$; $BF_{10} = 0.31$; Fig. 6f) before re-fixating the model. However, location errors

(i.e., placing an object at the wrong location) did not increase significantly in response to high distraction. We found no statistically significant differences in location error rates between distraction conditions when placing the first object ($\beta = -0.05$, $SE = 0.05$, $z = -1.04$, $p = 0.30$, $CI_{95\%} = [-0.14, 0.04]$; Fig. 6e), with a follow-up Bayesian $t$-test ($BF_{10} = 0.45$) providing no credible evidence in favour of the null hypothesis. During the second placement, there also was no detrimental effect of distraction on location errors. Instead, participants made more location errors when distraction was low ($\beta = 0.32$,

$SE = 0.14$, $z = 2.37$, $p = 0.02$, $CI_{95\%} = [0.06, 0.59]$; Fig. 6g). However, a follow-up Bayesian $t$-test ($BF_{10} = 2.16$) provided no credible evidence for the alternative hypothesis.

## Discussion

Using a multivariate VR approach, we discovered multifaceted consequences of visual distraction on several interconnected subcomponents of natural behaviour. We show that visual distraction while searching for task-relevant objects in the environment leads to a subtle change in subsequent sensorimnemonic decisions: when faced with high distraction, participants were more likely to (re-)encode information about upcoming task-relevant objects before search, compared to relying on information in memory.

Notably, the cumulative consequences of the focal effect of distraction on sensorimnemonic decisions affected both overall WM usage and encoding demands, ultimately changing how participants coordinated sampling information from the environment and guiding behaviour through information in memory. Specifically, participants used less WM, instead encoding information from the environment more often when faced with high distraction during visual search. These changes in the coordination of encoding, visual search, and WM usage had critical consequences for the overall task behaviour: participants made more errors, were slower to complete the task, and had to perform more body movements when encountering high distraction. Overall, our results illustrate that focal effects of visual distraction (i.e., distraction was introduced at a specific, isolated stage within the extended task) on memory-guided natural behaviour can have downstream, cascading consequences on processes that are contingent on the distraction-affected event[42–45,47].

From related research on visual distraction in standard laboratory tasks, we came to appreciate various proposed mechanisms of how distraction interacts with WM. While it has been argued that WM is remarkably robust in the face of distraction[35,61,62], distraction during the delay-period of WM tasks, both through perceptual input and interfering tasks, has been shown to interfere with WM performance[4,6,63–65] (for a review see ref. 35). Related, distraction has a more pronounced impact on WM performance when response demands to the distracting input increase (e.g., passive viewing vs. dual-task scenarios like visual searching while simultaneously holding information in WM)[3,9,66,67]. Further, it is well established that visual search is affected by target-distractor similarity[57,58] – which allowed us to reliably induce different levels of visual distraction during search in our VR paradigm. While a separate line of research has also explored the effects of visual search on concurrent WM, the precise mechanisms underlying how visual search disrupts WM remain a subject of ongoing debate[59–61].

Our results suggest that visual distraction can also compromise WM *usage* during natural behaviour, prompting participants to encode from the external environment more frequently. In our task, however, WM usage could have been impacted by both distraction interference (i.e., more frequent encoding served to compensate for such interference) and/or proactive shifts in the coordination of encoding and WM usage. A strong example of a proactive shift in WM usage can be found in participant's response to increased movement effort, prompting participants both to encode more objects and subsequentially decrease their encoding frequency through an increased reliance on WM (see ref. 42 for a more detailed discussion). In contrast, the change in the coordination of WM usage and encoding in response to distraction seems to have been reactive. Specifically, distraction did not lead to a systematic change in *how* participants encoded from the environment (i.e., the number of objects encoded, or time spent in each encoding episode) but only *how often* participant referred to the model – particularly when behaviour must be guided by information closely tied to the present distraction. In line with existing research on reactive distraction interference in WM[34,35], an increase in encoding frequency could serve to compensate for impaired content in WM. Alternatively, while using WM necessitates the appropriate content to be present, additional factors may influence the decision to act on it[68–70]. For instance, the prospect of a high distraction search could subtly influence participants' sensorimnemonic

decisions towards using less memory, without necessitating any substantial strategic changes in encoding behaviour or impaired WM content. Further research is required to distinguish between potential drivers of distraction-induced changes in the balance of WM encoding and usage.

A related question is how to interpret the overall minimal reliance on WM[41,42] across all experimental conditions – even when distraction was low. We propose that relying on WM less can carry strategic adaptive advantages by dynamically reducing distractor interference (in line with ref. 71). For example, when using a minimal memory strategy in our task, less information in WM needs to be selected for prioritisation and protected from external sources of distraction[35,72–75]. Keeping the number of items in WM at a minimum could additionally lessen internal inter-item competition between items held in WM concurrently[76–78]. In line with this proposal – and further strengthening the assumption that changes in WM usage can be associated with distraction interference instead of strategy changes – is the finding that distraction decreased WM usage more in conditions where WM is used more (i.e., high movement effort). Flexibility in the coordination of encoding and WM usage during natural behaviour therefore introduces opportunities to compensate for sources of distraction in our environment and mind. Such flexibility, however, is often not captured by highly controlled laboratory tasks used to uncover the mechanisms that help us handle distracting information. This highlights the importance of complementing traditional laboratory studies with more ecologically valid tasks in order to gain a fuller understanding of the consequences and mechanisms of distraction on cognition and behaviour.

### Limitations

Our findings bring the important realisation that effects of visual distraction on memory-guided natural behaviour can be rather focal, but even then, they can exert cascading consequences on processes that are contingent on the distraction-affected event. Importantly, we do not wish to claim that the particular pattern of consequences we observed holds for all forms of distraction. For instance, we only find evidence for distraction interference with identity-related sensorimnemonic decisions. Crucially, in the work reported here, we introduced a specific source of visual distraction: we increased the perceptual visual similarity between irrelevant objects and the encoded identity information – known to contribute to the effect of distraction[35,79] – but did not manipulate the similarity of location information. Distraction during natural behaviour, however, can take various forms: once we found the sugar, we might find ourselves in front of multiple mixing bowls, only some to which we need to add the sugar. In addition to visual distraction, other forms of distraction may also be present, such as auditory stimuli (e.g., hearing your dog bark)[80–83] or interruptions through task-irrelevant events (e.g., getting a call)[3,66,67,84]. Our study focused on one particularly powerful form of distractor interference and future research is necessary to uncover consequences of other types of distraction.

### Conclusion

Using an immersive VR approach, we discovered that effects of visual distraction on memory-guided naturally unfolding behaviour can be rather focal but nevertheless have downstream, cascading consequences on processes that are contingent on the distraction-affected event. Specifically, distraction influenced sensorimnemonic decisions; resulting in changes to the coordination of WM usage and encoding during extended behaviour, which ultimately slowed down behaviour and increased costly body movements. These results underscore the importance of considering complex behaviour with high external validity[85] to gain a comprehensive understanding of the consequences of distraction on cognition and behaviour. Here, we demonstrate the potential of using VR paradigms to generate rich, multivariate behavioural data to achieve these aims.

### Data availability

The post-processed data (segmented and summarised) are available at the Open Science Framework: https://osf.io/ze5p3/. Raw data is available from https://zenodo.org/records/11073009.

## Code availability

Code for all performed analyses is available at the Open Science Framework: https://osf.io/ze5p3/.

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

## Acknowledgements

This research was funded by a Wellcome Trust Senior Investigator Award to A.C.N. (104571/Z/14/Z) and a James S. McDonnell Foundation Understanding Human Cognition Collaborative Award (220020448). The Wellcome Centre for Integrative Neuroimaging is supported by core funding from the Wellcome Trust (203139/Z/16/Z and 203139/A/16/Z). This research is supported by the NIHR Oxford Health Biomedical Research Centre (NIHR203316). The views expressed are those of the author(s) and not

necessarily those of the NIHR or the Department of Health and Social Care. For the purpose of open access, the author has applied a CC BY public copyright licence to any Author Accepted Manuscript version arising from this submission. This work was additionally funded by the Deutsche Forschungsgemeinschaft (DFG, German Research Foundation)—project number 222641018—SFB/TRR 135, sub-project C7 and the Hessisches Ministerium für Wissenschaft und Kunst (HMWK; project 'The Adaptive Mind') to M.L.V. L.K. is supported by a studentship from the Clarendon Fund and the Department of Experimental Psychology, University of Oxford. The funders had no role in study design, data collection and analysis, decision to publish or preparation of the manuscript.

## Author contributions

Levi Kumle: Conceptualisation, Formal analysis, Investigation, Methodology, Software, and Writing – original draft and review & editing. Melissa L.-H. Vo: Supervision, Writing - review & editing. Anna C. Nobre: Conceptualisation, Supervision, Writing - review & editing. Dejan Draschkow: Conceptualisation, Formal analysis, Investigation, Methodology, and Writing – original draft and review & editing.

## Competing interests

The authors declare no competing interests.
