## [Peer Review File · Communications Psychology]

4th Jan 24

Dear Mx Kumle,

Thank you for your patience during the peer-review process. Your manuscript titled "Multifaceted consequences of distraction during natural behaviour" has now been seen by 3 reviewers, and I include their comments at the end of this message. They find your work of interest, but raised some important points. We are interested in the possibility of publishing your study in Communications Psychology, but would like to consider your responses to these concerns and assess a revised manuscript before we make a final decision on publication.

We therefore invite you to revise and resubmit your manuscript, along with a point-by-point response to the reviewers. Please highlight all changes in the manuscript text file.

The Reviewers raise various methodological concerns that need to be addressed; to address the issues satisfactorily, you will need to collect further data where additional analyses cannot entirely demonstrate methodological soundness and strength of evidence.

The key concerns include Reviewer #1's concerns regarding generalizability, Reviewer #2's concerns regarding the potential effect of changes in behavioural strategy, and Reviewer #3's concerns regarding temporal task effects.

Also, please ensure you provide clear justifications for all analyses and report statistics in full wherever they appear (including effect sizes and Confidence Intervals). More detailed guidance on our requirements for statistics reporting and interpretation are included in the attached checklist and we encourage you to follow Reviewer #2's suggestion of adding BFs to all analyses, not only the null results (for which positive evidence derived from BFs or equivalence tests is a journal requirement).

I have attached a checklist for our formatting and editorial policies, please ensure your revision complies with the checklist.

Please use the following link to submit your revised manuscript, point-by-point response to the referees' comments (which should be in a separate document to any cover letter) and the completed checklist:

[link redacted]

Please do not hesitate to contact me if you have any questions or would like to discuss these revisions further. We look forward to seeing the revised manuscript and thank you for the opportunity to review your work.

Best regards,

Antonia Eisenkoeck

Antonia Eisenkoeck

Senior Editor

Communications Psychology

* **TRANSPARENT PEER REVIEW:** Communications Psychology uses a transparent peer review system. This means that we publish the editorial decision letters including Reviewers' comments to the authors and the author rebuttal letters online as a supplementary peer review file. However, on author request, confidential information and data can be removed from the published reviewer reports and rebuttal

letters prior to publication. If your manuscript has been previously reviewed at another journal, those Reviewers' comments would not form part of the published peer review file.

REVIEWERS' EXPERTISE:

Reviewer #1: working memory, virtual reality

Reviewer #2: visuo-motor coordination

Reviewer #3: working memory, visuo-motor coordination

REVIEWERS' COMMENTS:

Reviewer #1 (Remarks to the Author):

Kumle et al. developed a new study protocol using virtual reality, which allowed for investigation of perceptual and cognitive processes as they unfolded during naturalistic behaviour. Specifically, participants were asked to copy a two-dimensional array of objects within a virtual environment by visually learning the model array, searching and selecting target objects in a separate "resource pool", and then placing them in appropriate cells of a "workplace". The authors manipulated the degree of visual distraction by making objects in the resource pool more or less visible, and also the amount of movement the task entailed by placing the model in front or to the side of the workplace. They evaluated participants' performance by separately measuring its subcomponents (encoding, visual search, working memory usage, and sensory-mnemonic decisions) as well as overall behaviour (the time it took to complete the task and the number of head movements made). Generally, results showed that distraction affected many of these measures, capturing the inter-related nature of these subcomponents in naturally occurring behaviour.

The manuscript is very well written and it did a good job in demonstrating the methodological advantage of the study protocol - it can investigate human behaviour and cognition in an externally and ecologically valid fashion while maintaining a good degree of internal validity. I think highly of the manuscript from this point of view. At the same time, however, I also think the strength of the manuscript is largely limited to this aspect. As noted by the authors themselves (p. 9, lines 28–37), some of the findings may well be rather specific to the particular set-up of the task and/or the virtual environment (e.g., location-

related sensory-mnemonic decisions being unaffected by distraction - this was most likely a mere consequence of creating distraction in the resource pool, not in the model or workplace where location information was most relevant). I also remain unconvinced by other empirical claims made by the authors, as described below. Taken together, I feel ambivalent about this manuscript - it surely is a high-quality manuscript, but I am not certain whether it will eventually make a strong empirical paper.

Major issues

- On p. 7, while describing identity-related sensory-mnemonic decisions, the authors stated, "encountering high distraction led to participants deciding to rely on memory less. When faced with high distraction, participants were less likely to decide to pick up a second object [...], which would have required behaviour to be guided by using a second identity attribute from memory" (lines 32–35). I am not sure if this result should be characterised as memory-driven because in the low distraction condition, the resource pool itself helped participants identify target objects, decreasing the need of looking back at the model for confirming the identify of targets. In this interpretation, the difference between low and high distraction conditions was driven externally by visual properties of the resource pool - i.e., it was sensory-driven.

- More generally, the pictorial difference in the resource pool between low and high distraction conditions could have induced a strategy difference, potentially creating differential task demands in the two conditions. Only in the low distraction condition, participants had the option of relying primarily on the resource pool to learn target objects. For example, instead of doing visual search on the resource pool, they can simply start at one corner of the resource pool and select the nearest target object, and then search for that object within the model to learn its location. Then they go back to the resource pool, select the next target object, and repeat the same. In this approach, participants can systematically go through target objects in the resource pool (e.g. going row-by-row or column-by-column), largely removing the need of doing visual search on the resource pool. One might argue that this was indeed a valid consequence of having high vs. low distraction; to me, it was more of an artefact caused by the particular way of implementing the two levels of distraction.

- Some of the differences between conditions were significant but numerically very small - e.g., those shown in Fig. 5B. In addition to evaluating their statistical reliability, they should also be interpreted in terms of their effect size.

Minor/specific issues

- I found the term "locomotive effort" somewhat misleading because in the 90° condition, participants did not have to locomote from the workplace to the model - all it took was to turn their head and body

while remaining in front of the workplace. I guess this term may be preferred for consistency with the authors' other papers, but I just can't help but feel there should be a better name for it.

- On p. 7, line 26, the authors refer to Fig. 4B, but I think it should have been Fig. 5B.

- On p. 13, line 28, HMD is misspelled as "HDM".

- Two styles of in-text citations - Nature's superscript numbers and APA's (Author, year) format - are mixed in the manuscript.

Reviewer #2 (Remarks to the Author):

In this paper, the authors developed a novel VR paradigm to investigate the impact of distractions on various cognitive aspects of natural behaviors. These components include encoding, visual search, working memory utilization, and decision-making. By adjusting the opacity of distractors in a core visual search task, they unveiled nuanced effects of distraction across these cognitive dimensions. Intense distraction resulted in more frequent encodings, prolonged visual searches for target information, reduced the number of attributes employed in working memory, and heightened the necessity to re-encode external information to successfully complete the task.

The study is strongly motivated, spanning various subfields of human cognition. The paper is meticulously organized and well-written. The authors' innovative approach to addressing the crucial question about the impact of distraction on natural behaviors has yielded valuable insights. The analyses conducted were both rigorous and thorough. I particularly commend the authors for their efforts in decomposing complex natural data into subcomponents that relate to specific cognitive processes, a strategy that I think is instrumental in revealing fundamental aspects of distraction disruptions on task performances. However, I have three major concerns regarding the authors' interpretation of the data and a couple of additional points of concern.

Major concerns:

1. One main conclusion drawn in the paper is that distraction "altered the coordination of working memory (WM) usage and encoding during extended behavior," derived from observations of reduced WM usage and increased encoding frequency in the high distraction condition compared to low

distraction conditions. The authors claimed that this change represents a proactive and adaptive response to avoid interference. While this conclusion is intriguing, it requires further evidence. An alternative explanation for the observed pattern is that distractions compromised WM capacity (both in quantity and quality) and therefore WM usage, prompting participants to encode target information more frequently to complete the task. According to this interference account, the change in the coordination of WM usage and encoding is reactive, serving to compensate for distractor interferences.

This interference account is not only more intuitive but also aligns better with the data. If participants were proactively employing a strategy to rely less on WM and encode more frequently in the high-distraction condition, one would expect to observe fewer encoded items and less time spent in each encoding episode, coupled with an increase in encoding frequency. However, the data presented by the authors (Fig 3 and Sfig 1) show no change in the number of items encoded or time spent in each encoding episode but an increased encoding frequency, consistent with the interference account.

If the authors aim to assert that there is a strategic change between high and low distraction conditions, additional evidence is necessary. For instance, examining differences between early and later high distraction trials could shed light on the development of the strategy over the course of the task. Additionally, exploring individual differences in strategies could provide further insights.

2. Besides expected and exciting main result patterns, the authors found a couple unexpected results but did not address them sufficiently. Here are two examples. On page 4, the authors found an interaction between distraction and locomotive effort in model viewing time. Participants encoded slightly longer when distraction was high compared to low during the low locomotive effort condition. On page 7, there was an interaction between distraction and locomotive effort. Distraction decreased WM usage more when locomotive effort was high. Though some of those results might not be of key interest to the authors' main conclusion, it would be worthwhile to discuss and interpret this finding. For example, the observation that distraction decreased WM usage more when locomotive effort was high seems to suggest that distraction interference was larger in conditions where WM is more used than when it is not. This is additional evidence suggesting that changes in WM usage can be associated with distraction interference instead of strategy changes, which have been neglected across the text.

3. The finding that distractions primarily induced identity errors rather than location errors is intriguing, but I found it a bit confusing as to how these errors were separated. Identity errors would remain independent of location errors, considering that pickups occurred before placements. However, location errors could be contaminated by identity errors. When an incorrect item was picked up, the subsequent placement would also be inaccurate. I am curious whether the authors analyzed only trials in which correct items were picked up to assess location errors, or if they overlooked the distinction between correct and incorrect pickups. The latter scenario would imply that location errors might have been influenced by identity errors. It is necessary for the method section to explicitly address this concern, offering clarification on how the analysis was done.

Minor points:

- 1 Bayes factors (BF) should be reported for all analyses.
- 2 same for effect sizes
- 3 In fig 6, the y axis for all identity error figures should be aligned.
- 4 In all figure legends, it should be specified what *, **, *** mean.

Reviewer #3 (Remarks to the Author):

The authors investigate the effects of visual distraction and locomotor effort on different components of an object copying task. They conclude that visual distraction interfered with encoding, visual search, working memory reliance, and decision-making. Higher locomotive effort also increased working memory reliance. I think the research question is interesting and fits nicely within the scope of the journal. The methodology is clearly explained, and the figures are nicely presented (I particularly appreciate the plotting of individual data points). However, I do have several queries regarding the authors' choice of terminology, and rationale for the experimental design and statistical analyses that I elaborate on below. I think these points need addressing in a revision.

Major

1. I find the use of the term 'distraction' throughout the paper to be rather misleading and confusing. When I read the title, I expected the experiment to focus on 'distractions' in the sense of someone interrupting the naturalistic task currently being undertaken (e.g., the phone rings during the baking example that the authors give). However, it becomes apparent that the authors are referring to, and manipulate, characteristics of visual distractors. I think the narrative of the paper and terminology used could be adjusted to make this clearer for the reader. Below are some examples:
 - a. Results Section (Page 3, line 23), 'We first consider the consequences of distraction on summary measures...'. This sentence could be interpreted as 'distraction' referring to both the independent variables collectively (i.e., visual distractors and locomotor effort could both be distractions) or as referring to the effect of visual distraction variable, specifically.
 - b. The sub-headings of the results section all highlight 'Distraction does...'. In line with my above comment, the use of this general term 'distraction' initially made me think the authors were referring to both variables collectively, but they are not. Rather, there is much more emphasis on the visual distraction effect compared with the locomotor effort effect.

c. Discussion (Page 9, line 16). 'Using a novel VR method, we discovered dissociable and multifaceted consequences of distraction on several subcomponents of natural behaviour'. This implies 'distraction' is a multifaceted construct but in this experiment, it is specifically the opacity of a distractors.

d. More generally, the authors swap between using 'visual distraction' and just 'distraction' to refer to the same thing. I think the paper would be clearer if consistent terminology was used throughout.

2. The authors explain that the rationale for including locomotor effort as an independent variable was to 'induce more variability in participants' WM usage and investigate how the impact of distraction varies depending on locomotor demands (page 2, lines 47 – 50).' From my understanding, the authors are interested in the interaction between visual distraction and locomotor effort. However, I miss the rationale explaining why there would be an interaction because they effect different subcomponents of the task. Specifically, visual distraction effects encoding and search whereas locomotion effects WM usage. I would appreciate some additional clarity for the rationale behind including locomotor effort as an independent variable, and the logic behind expecting an interaction.

3. Following on from the above, I would appreciate the rationale behind some of the statistical analyses performed and clarity on what hypotheses are being tested. From my understanding, the authors expect the two different independent variables to have an effect on different subcomponents of the task. Therefore, there are several analyses reported in the paper that I am not convinced are testing well-motivated hypotheses. For example, the effect of locomotive effort on display completion time (Page 4, lines 6-11) and the effect of locomotive effort on visual search time (Page 6, lines 2- 8).

4. Results: 'Distraction reduces memory usage'. My interpretation is that the most compelling effect in Figure 5b is that of locomotor effort on memory usage, yet the authors don't really comment on this. Given the focus of the paper on naturalistic tasks, it seems more relevant that rotating 90 degrees has such a clear effect on working memory usage. In my opinion, the effect of distraction is less compelling, especially when you consider that only integer values are possible for the number of attributes measure (although there is consistency in the downward slope of the participant level data, these are mostly within an single attribute value). I wonder whether the authors think that participants are doing something meaningfully different in the high compared with low distraction condition or are they mostly using the same number of attributes? I do think this is a nice measure and would be interested to see the distribution of attributes in WM, which would help address my query above.

5. The section on sensory-mnemonic decisions is interesting and highlights the sequential nature of this task. I think an important consideration is how the participants' behaviour changes over the course of the task because of the task itself (i.e., rather than because of the visual distraction). My understanding is that all aspects of the task become easier as the participant moves more blocks from the resource pool to the workspace because there are less items to search in the resource pool, and less possible locations to place the item in the workspace. I wonder whether the authors have thought about the implications

of this are for their results. For example, I would expect the attributes in WM measure to increase over time (less reliance on the model) because there is less need to verify the correct location for an item once the workspace becomes fuller (i.e., knowing an item is roughly top left becomes sufficient rather than needing to know the position is [1,2] in the array). I think a visualisation showing how some of the measures change over the course of the task would benefit the manuscript because this seems an important consideration in the task.

6. Overall, I find the discussion rather difficult to follow. The authors make claims about 'dissociable', 'multifaceted', 'cascading' consequences of distraction but I struggle to understand what they really mean by these terms. Specifically:

a. What is multifaceted? The 'distraction' investigated here is very specific (i.e., visual distraction manipulated by changing opacity).

b. What consequences of distraction are dissociated? Do the authors mean there is a different consequence of distraction in the different parts of the task.

c. What does cascading refer to? This is almost contradictory to 'dissociating'. It implies that the effect of visual distraction in one component of the task has an effect on another.

I would appreciate the authors providing a clearer narrative throughout the discussion and suggest considering using different terminology.

Minor

1. Abstract. 'Interfered with' is rather vague. Can this be more specific by referring to the DV (e.g., a decrease in WM load).

2. 'HDM' should be 'HMD' in a few places (e.g., page 11, line 36)

3. Figure 1a. In this figure, the resource pool is 3 rows*8 columns. It would be nice if this were consistent with the rest of the manuscript (both figures and text).

4. Page 12, line 28. 'Participants could not pick up another object until their mistake was corrected.' To what extent does this part of the experiment constrain the decision process and what are the implications of this for your results?

5. Frame durations. What does this mean? Does each frame not have the same duration (90 Hz)?

6. I think 'locomotive effort' could be better replaced with 'locomotor effort' throughout the manuscript because it's the participant that moves.

Dear Reviewers,

Thank you for the exceptionally high quality of reviews. They were fair, thorough, and to the point. Before turning to our point-by-point replies, we highlight and clarify several general points.

First, the central claim of our work is that effects of visual distraction on memory-guided natural behaviour can be rather focal but nevertheless have cascading consequences. To reach this insight, we developed an innovative methodological approach that segments complex behaviour into cognitive subcomponents and allows us to trace the consequences of distraction all the way from overall task performance to the decision-making processes that gate memory usage. We do not wish to claim that our take-aways are true for all forms of distraction (**generalisability**), as our experimental approach was tailored to demonstrate this fundamental principle by manipulating one specific type of distraction (visual identity-based distraction).

Second, we now provide analyses to test for any potential effects of **changes in behavioural strategy**, providing additional evidence that the effects of distraction are focal and do not lead to major changes in behavioural strategy. Nevertheless, many central cognitive operations are affected by the visual distraction introduced in our experimental design.

Third, we have conducted control analyses that demonstrate that our experimental effects go above and beyond any **temporal task effects**. We have included the relevant analyses in the response letter and manuscript.

Fourth, we originally named our secondary experimental manipulation “locomotive effort” to be consistent with our previous work. The reviewers rightly point out that this could lead to some ambiguous interpretations. Because the experimental manipulation made participants *move* their entire body, we have now renamed it “movement effort”.

Finally, we wish to thank the reviewers for highlighting the novelty of our work, that it spans various subfields of human cognition, and the importance of our approach for beginning to understand visual distraction during natural behaviour.

We again wish to thank you for your time and attention.

Sincerely,

Levi Kumle, Melissa L.-H. Võ, Anna C. Nobre & Dejan Draschkow

Department of Experimental Psychology & Oxford Centre for Human Brain Activity, Wellcome Centre for Integrative Neuroimaging, University of Oxford, UK

REVIEWERS' EXPERTISE:

Reviewer #1: working memory, virtual reality

Reviewer #2: visuo-motor coordination

Reviewer #3: working memory, visuo-motor coordination

Reviewer #1 (Remarks to the Author):

Kumle et al. developed a new study protocol using virtual reality, which allowed for investigation of perceptual and cognitive processes as they unfolded during naturalistic behaviour. Specifically, participants were asked to copy a two-dimensional array of objects within a virtual environment by visually learning the model array, searching and selecting target objects in a separate "resource pool", and then placing them in appropriate cells of a "workplace". The authors manipulated the degree of visual distraction by making objects in the resource pool more or less visible, and also the amount of movement the task entailed by placing the model in front or to the side of the workplace. They evaluated participants' performance by separately measuring its subcomponents (encoding, visual search, working memory usage, and sensory-mnemonic decisions) as well as overall behaviour (the time it took to complete the task and the number of head movements made). Generally, results showed that distraction affected many of these measures, capturing the inter-related nature of these subcomponents in naturally occurring behaviour.

The manuscript is very well written and it did a good job in demonstrating the methodological advantage of the study protocol - it can investigate human behaviour and cognition in an externally and ecologically valid fashion while maintaining a good degree of internal validity. I think highly of the manuscript from this point of view. At the same time, however, I also think the strength of the manuscript is largely limited to this aspect. As noted by the authors themselves (p. 9, lines 28–37), some of the findings may well be rather specific to the particular set-up of the task and/or the virtual environment (e.g., location-related sensory-mnemonic decisions being unaffected by distraction - this was most likely a mere consequence of creating distraction in the resource pool, not in the model or workplace where location information was most relevant). I also remain unconvinced by other empirical claims made by the authors, as described below. Taken together, I feel ambivalent about this manuscript - it surely is a high-quality manuscript, but I am not certain whether it will eventually make a strong empirical paper.

We thank the reviewer for this candid appraisal and for recognizing and highlighting the methodological strengths of our manuscript. We understand how the reviewer may have concluded that the specificity of our effects diminished the empirical impact of the paper, and we take responsibility for not making the general implications of our findings much clearer.

We manipulated a specific type of distraction within a well validated immersive (working memory) framework with which we have developed expertise in order to make two contributions: (1) develop analytical tools that segment continuous behaviour into its core subcomponents and (2) to *illustrate* how specific perturbations in sensory parameters of the experimental setting can have multiple cascading consequences to behaviour depending on the cognitive subcomponent affected. So, although the pattern of effects is, sensibly, specific to the experimental framework and task, it is illustrative of the important and generalisable principle that focal manipulations can have various and interacting consequences for other components of behaviour, as well as overall performance.

To address the important shortcoming in our communication noted by the reviewer, we have made the intent and implications of our study much clearer. We sharpened our title, fine-tuned

statements about this throughout our manuscript, and substantially restructured our discussion. We hope that our work will be the first step towards a more comprehensive investigation of the consequences of *many different forms* of distraction on memory-guided natural behaviour. We respond to specific reviewer concerns below.

Major issues

To provide a more coherent narrative in our response, we swapped the order of major concern 1 and 2.

- More generally, the pictorial difference in the resource pool between low and high distraction conditions could have induced a strategy difference, potentially creating differential task demands in the two conditions. Only in the low distraction condition, participants had the option of relying primarily on the resource pool to learn target objects. For example, instead of doing visual search on the resource pool, they can simply start at one corner of the resource pool and select the nearest target object, and then search for that object within the model to learn its location. Then they go back to the resource pool, select the next target object, and repeat the same. In this approach, participants can systematically go through target objects in the resource pool (e.g. going row-by-row or column-by-column), largely removing the need of doing visual search on the resource pool. One might argue that this was indeed a valid consequence of having high vs. low distraction; to me, it was more of an artefact caused by the particular way of implementing the two levels of distraction.

This is a very interesting point. Overall, the flexibility in self-structuring behaviour during the task did allow for potential strategy changes and we believe that any type of strategy shift would be an extremely interesting consequence of distraction. We acknowledge that we neglected to adequately discuss any such potential changes in overall task-performance strategy (also see reviewer #2. major comments 1 & 2). Importantly, large-scale strategic shifts would present themselves through distinct patterns, visible in our dense behavioural metrics.

The alternative strategy described by the reviewer, for instance, would change the function of the model and resource pool: a “*Resource-as-reference*” instead of “*Model-as-reference*”-strategy. When following the proposed “*Resource-as-reference*” strategy, participants would only need to encode the location information from the model display. Accordingly, participants would first pick up a target from the resource pool, and then – *with the target object already in hand* – turn to the model to find and encode the location of the already grabbed target. After placing the target, participants would turn back to the resource pool, pick up the next target and then again, turn to the model with a target object grabbed. What we would not see in this kind of strategy, however, is that participants predominantly look at the model without an object in their hand before turning to the resource table to pick up the next object.

First, all analysis on sensory-mnemonic decisions in the main text (Fig. 6A) only included behavioural sequences that started with encoding from the model *without* a target in hand (overall, 70 % of sequences in 90° movement effort condition started with encoding without an object in hand), making it unlikely that we missed the specific change proposed by the reviewer in our interpretation of the results. We highlight this point in the Methods section (p. 9, line 23).

We now additionally report analyses testing for potential strategy differences on p. 14, line 1 and in the Supplementary Notes 3. Note that within these analyses, the “*Resource-as-reference*”-strategy suggested by the reviewer should result in a large increase of “*Placement-first*” sequences in the low distraction condition, as these are sequences where participants encoded

from them model with an object already in hand. In contrast, we do not see a systematic change in task-performance strategies, suggesting that participants did not follow a “Resource-as-reference”-strategy when distraction was low.

Main manuscript, p.14, line 1:

“Importantly, the flexibility in self-structuring behaviour during the task did additionally allow for potential strategy changes. That is, behavioural sequences could have a different structure, even if participants are using the same number of attributes in WM. For example, using two attributes in WM could be the result of first searching for and picking up a target object (i.e., identity feature used) and then directly placing this object (i.e., location attribute used). At the same time, participants could have used two attributes in WM by first placing an already picked-up object (i.e., participant encoded while holding an object in hand) before continuing to search for and pick-up another object. However, while visual distraction affected WM usage, it did not lead to major strategy changes (see Supplementary Notes 3).”

Supplementary Notes 3, p. 4 in Supplementary Information:

*“**Analysis.** We first determined different sequence types by dividing behavioural sequences according to the number of attributes used within them, as well whether they started with search and pick-up (i.e., “search and pick-up first” sequences, **Fig. S3A**) or placement (i.e., “Placement first” sequence, **Fig. S3A**). We then computed the probability of different sequence types. Differences in means between distraction conditions were analysed nested in movement effort conditions, using paired pairwise t-tests as well as paired Bayesian t-tests. All pairwise comparisons are reported in **Table SA** below.*

***Results.** Overall, we do not find evidence for systematic strategy shifts induced by visual distraction (see **Fig. S3C** and **Table SA**). While we do find differences between distraction conditions in some sequence types (e.g., S1 in 90° movement effort), overall patterns were unaffected and mean differences were small. Note, that the results here are a post-hoc decomposition of the differences found in the main analysis of WM usage (see **Fig. 5C**). “*

Figure S3. Flexibility in self-structuring behaviour introduces possibilities for strategy changes. *a)* Behavioural sequences could have a different structure, even if participants are using the same number of attributes in WM. That is, sequences could start with search and pick-up (i.e., “Search and pick-up first”-sequences; sequence starts without object in hand) or placement right after encoding (i.e., “Placement first”-sequences; sequence starts with object in hand). *b)* Illustration of the structure of different sequence types (in 90° movement effort condition). *c)* Probability of using different sequence types. S indicates sequences that started with search, P indicates sequences that started with placement. Error bars depict standard error of the mean (N = 30). The symbols *, **, and *** in the figure denote statistical significance with p-values less than 0.05, 0.01, and 0.001, respectively.

To probe further whether any systematic strategy shifts occurred, specifically in line with a “Resource-as-reference”-strategy, we conducted a set of descriptive post-hoc analyses in this response letter. A central implication of the alternatively proposed “Resource-as-reference”-strategy is that participants would structure their object copying behaviour in accordance with the resource pool. This could be achieved in various ways, either starting at one of the corners/rows/columns and systematically moving through the resource table, or even moving through the resource pool in an unsystematic way but nevertheless keeping track of the progress in regard to the resource pool. Critically, any “Resource-as-reference” strategy should prevent that the order of pick-ups and placements could be predicted based on the location of and distance between targets in the model display, given that target locations in the resource and model were entirely unrelated.

We again found no evidence suggesting a systematic strategy shift. In contrast, we found a strong pattern showing that – in both distraction conditions – the order of pick-ups and placements followed the location of and distance between targets in the model display (i.e., strongly suggesting a “Model-as-reference”-strategy). Specifically, participants showed a strong

tendency to start in the upper left corner of the model display, systematically working through to the lower right corner (**Fig R1**) while preferring to successively place targets that are close together in the model (**Fig R2**).

Overall, while we cannot entirely rule out the possibility that, in isolated instances, participants may have used different strategies, we find no evidence for major and consistent strategy changes caused by our distraction manipulation. Instead, we provide converging evidence that the effects of distraction are small and focal, but nevertheless affect many central cognitive operations within a common task-performance strategy.

Figure R1. Descriptive analysis of the spatial-temporal order of selected targets in relation to the model display. A) Model display with numbered slots as reference. In all conditions, we see a strong tendency to start in the upper left corner of the model display (slot 1), then systematically working through to the lower right corner (slot 18). In contrast, a shift towards a “Resource-as-reference”-strategy would result in the probabilities being closer to/at chance in the low distraction

condition only. **B) Probability of three example slots being selected for copying throughout the trial.** To illustrate this pattern, we first calculated the probability of a target presented in a given model slot to be selected for copying at a given time throughout the trial (i.e., as indexed by the number of targets (out of 8) already placed). Targets presented in model slot 1 showed a very strong tendency to be copied first (top, dotted line represents chance). Targets shown in slot 10 (middle), were most likely copied in the middle of the trial. Targets shown in slot 18 (bottom) were more likely to be copied at the end of the trial. Importantly, this pattern is visible in both distraction conditions. **C) Difference to chance of the probability of all slots being selected for copying throughout the trial.** Extending the description presented in B), we illustrate the probability of all slots (baselined at chance) of a target being selected for copying at a given time in the trial. Dotted lines represent divisions between rows in the model. Overall, we find a comparable pattern in all conditions. Participants tended to start in the upper left corner, then working through the model in a left-to-right order, ending at the bottom right corner. This indicates that participants consistently followed a “Model-as-reference”-strategy in all experimental conditions.

Figure R2. Distance between consecutive placements in reference to the model display. We additionally computed the Manhattan-distance between model slots that were selected for copying consecutively (i.e., distance between slot 1 and 2 equals 1, distance between slot 1 and 9 equals 3). We additionally computed chance performance (as implied by the “Resource-as-reference”-strategy) by computing the distance between target slots in a random order. Overall, we see a strong tendency to consecutively place target objects that are close to each other in the model display in all conditions.

- On p. 7, while describing identity-related sensory-mnemonic decisions, the authors stated, "encountering high distraction led to participants deciding to rely on memory less. When faced with high distraction, participants were less likely to decide to pick up a second object [...], which would have required behaviour to be guided by using a second identity attribute from memory" (lines 32–35). I am not sure if this result should be characterised as memory-driven because in the low distraction condition, the resource pool itself helped participants identify target objects, decreasing the need of looking back at the model for confirming the identify of targets. In this interpretation, the difference between low and high distraction conditions was driven externally by visual properties of the resource pool - i.e., it was sensory-driven.

This is a good point and tightly linked to the reviewer’s previous point. Above, we have demonstrated that no major strategy shifts occurred, and that participants predominant strategy was to first encode objects in the model and keep them in memory, then grab them in the resource pool, before placing them in the workspace. Depending on the exact time-point in a copying sequence, we could infer if the current behaviourally relevant attribute was an identity

(i.e., before search and pick-up) or location (i.e., before placement) attribute. Additionally tracking when participants chose to encode information from the model display therefore allowed us to quantify the trade-off between using sensory vs memory information.

However, the point raised by the reviewer nicely captures the interlinked and complex nature of consequences of distraction during natural behaviour – which are at the heart of the presented manuscript. Specifically, while we introduced sensory-driven differences in the resource pool (visual search subcomponent of the task), we were interested in the consequences on subcomponents (such as memory usage) that are dependent on these sensory-driven differences.

We acknowledge that our Discussion failed to do justice to the complexity of potential drivers of the distraction effect (also noted by reviewer #2; major concern 1 and 2). We therefore considerably restructured our discussion, now explicitly naming and discussing these potential drivers of differences between distraction conditions in memory-guided behaviour (p. 17, line 32):

“Our results suggest that visual distraction can also compromise WM usage during natural behaviour, prompting participants to encode from the external environment more frequently. In our task, however, WM usage could have been impacted by both distraction interference (i.e., more frequent encoding served to compensate for such interference) and/or proactive shifts in the coordination of encoding and WM usage. A strong example for a proactive shift in WM usage can be found in participant’s response to increased movement effort, prompting participants to both encode more objects and subsequently decrease their encoding frequency through an increased reliance on WM (see ⁴² for a more detailed discussion). In contrast, the change in the coordination of WM usage and encoding in response to distraction seems to have been reactive. Specifically, distraction did not lead to a systematic change in how participants encoded from the environment (i.e., the number of objects encoded, or time spent in each encoding episode) but only how often participant referred to the model – particularly when behaviour must be guided by information closely tied to the present distraction. In line with existing research on distraction interference in WM^{34,35}, these reactive changes could be caused by interfered WM content, with an increase in encoding frequency serving to compensate for such interference. Alternatively, while using WM necessitates the appropriate content to be present, additional factors may influence the decision to act on it ⁶⁷⁻⁶⁹. For instance, the prospect of a high distraction search could subtly influence participants’ sensory-mnemonic decisions towards using less memory, without necessitating any substantial strategic changes in encoding behaviour or interfered WM content. Further research is required to distinguish between potential drivers of distraction-induced changes to the coordination of WM usage and encoding.
“

- Some of the differences between conditions were significant but numerically very small - e.g., those shown in Fig. 5B. In addition to evaluating their statistical reliability, they should also be interpreted in terms of their effect size.

Thank you for catching this. Highlighting that distraction does not seem to affect certain subcomponents of behaviour to the same extent as movement effort is an important part of the results. We now additionally report effect sizes for all analyses.

Further, we now also contextualise our results, as exemplified by the results presented in Figure 5B. We additionally added a panel to Figure 5 and corresponding analyses (see reviewer #3, major

comment 4) which further highlight that the distraction-induced difference in WM usage was small, but reliable (p 13, line 32):

“Distraction also critically influenced the coordination between gathering information from the environment and using information in WM, although to a lesser extent. [...] Comparing the probability of using different numbers of attributes in WM (Fig. 5C) further highlighted that the distraction-induced difference in WM usage was subtle (see Supplementary information, Table 9 for a full reporting of the pairwise comparisons).”

Minor/specific issues

- I found the term "locomotive effort" somewhat misleading because in the 90° condition, participants did not have to locomote from the workplace to the model - all it took was to turn their head and body while remaining in front of the workplace. I guess this term may be preferred for consistency with the authors' other papers, but I just can't help but feel there should be a better name for it.

We originally named our secondary experimental manipulation “locomotive effort” to be consistent with our previous work. We agree with the reviewer that this could lead to some ambiguous interpretations. Because the experimental manipulation made participants *move* their entire body, we have now renamed it “movement effort”.

- On p. 7, line 26, the authors refer to Fig. 4B, but I think it should have been Fig. 5B. We apologize for the lack of clarity in our reference to Fig. 4B, which was intended to refer to the results regarding the overall reliance on WM in the two movement effort conditions. We now make this more explicit (p. 15, line 13):

“To increase sensitivity, we focused our analysis on trials with high movement effort, as participants predominantly relied on only one attribute when movement effort was low (see overall WM usage depicted in Fig. 4B), precluding us from observing longer sequences of memory-guided behaviour.”

- On p. 13, line 28, HMD is misspelled as "HDM".

Thanks for spotting this mistake! We've corrected all abbreviations of HMD.

- Two styles of in-text citations - Nature's superscript numbers and APA's (Author, year) format - are mixed in the manuscript.

Many thanks for bringing this inconsistency to our awareness! All citations are now in Nature's superscript format.

Reviewer #2 (Remarks to the Author):

In this paper, the authors developed a novel VR paradigm to investigate the impact of distractions on various cognitive aspects of natural behaviors. These components include encoding, visual search, working memory utilization, and decision-making. By adjusting the opacity of distractors in a core visual search task, they unveiled nuanced effects of distraction across these cognitive dimensions. Intense distraction resulted in more frequent encodings, prolonged visual searches for target information, reduced the number of attributes employed in working memory, and heightened the necessity to re-encode external information to successfully complete the task.

The study is strongly motivated, spanning various subfields of human cognition. The paper is meticulously organized and well-written. The authors' innovative approach to addressing the crucial question about the impact of distraction on natural behaviors has yielded valuable insights. The analyses conducted were both rigorous and thorough. I particularly commend the authors for their efforts in decomposing complex natural data into subcomponents that relate to specific cognitive processes, a strategy that I think is instrumental in revealing fundamental aspects of distraction disruptions on task performances. However, I have three major concerns regarding the authors' interpretation of the data and a couple of additional points of concern.

We thank the reviewer for the thorough review and positive appreciation of our work. The comments and suggestions of the reviewer allowed us to improve the clarity of several of our main conclusions. We will elaborate in our point-by-point responses below.

Major concerns:

1. One main conclusion drawn in the paper is that distraction "altered the coordination of working memory (WM) usage and encoding during extended behavior," derived from observations of reduced WM usage and increased encoding frequency in the high distraction condition compared to low distraction conditions. The authors claimed that this change represents a proactive and adaptive response to avoid interference.

While this conclusion is intriguing, it requires further evidence. An alternative explanation for the observed pattern is that distractions compromised WM capacity (both in quantity and quality) and therefore WM usage, prompting participants to encode target information more frequently to complete the task. According to this interference account, the change in the coordination of WM usage and encoding is reactive, serving to compensate for distractor interferences.

This interference account is not only more intuitive but also aligns better with the data. If participants were proactively employing a strategy to rely less on WM and encode more frequently in the high-distraction condition, one would expect to observe fewer encoded items and less time spent in each encoding episode, coupled with an increase in encoding frequency. However, the data presented by the authors (Fig 3 and Sfig 1) show no change in the number of items encoded or time spent in each encoding episode but an increased encoding frequency, consistent with the interference account.

If the authors aim to assert that there is a strategic change between high and low distraction conditions, additional evidence is necessary. For instance, examining differences between early and later high distraction trials could shed light on the development of the strategy over the course of the task. Additionally, exploring individual differences in strategies could provide further insights.

We appreciate the important comment provided by the reviewer and we apologise that our Discussion caused a misunderstanding. We completely agree that we currently lack evidence to

assert that there are strategic, proactive changes between distraction conditions and did not mean to cast the results in this light.

When discussing potential *proactive* responses to interference, we were instead referring to the *overall* low reliance on WM in *all* conditions. Specifically, the extent to which distraction can interfere with memory-guided behaviour is linked to how much participants are relying on memory to begin with. For example, when using a very minimal memory strategy in our task (i.e., using only one attribute in WM), participants can reduce distractor interference during target search. In this case, less information in WM needs to be protected from distraction. We therefore originally aimed to highlight that the overall tendency of low WM reliance during natural behaviour (also discussed in e.g., Ballard et al., 1995; Draschkow et al., 2021) could serve adaptive benefits in terms of distraction interference.

To clarify this, we have now added the following paragraphs to our Discussion:

First, we now explicitly discuss our results in light of both distraction interference and proactive strategy changes (p 17, line 32):

“Our results suggest that visual distraction can also compromise WM usage during natural behaviour, prompting participants to encode from the external environment more frequently. In our task, however, WM usage could have been impacted by both distraction interference (i.e., more frequent encoding served to compensate for such interference) and/or proactive shifts in the coordination of encoding and WM usage. A strong example for a proactive shift in WM usage can be found in participant’s response to increased movement effort, prompting participants both to encode more objects and subsequently decrease their encoding frequency through an increased reliance on WM (see ⁴² for a more detailed discussion). In contrast, the change in the coordination of WM usage and encoding in response to distraction seems to have been reactive. Specifically, distraction did not lead to a systematic change in how participants encoded from the environment (i.e., the number of objects encoded, or time spent in each encoding episode) but only how often participant referred to the model – particularly when behaviour must be guided by information closely tied to the present distraction. In line with existing research on distraction interference in WM^{34,35}, these reactive changes could be caused by interfered WM content, with an increase in encoding frequency serving to compensate for such interference. Alternatively, while using WM necessitates the appropriate content to be present, additional factors may influence the decision to act on it ⁶⁷⁻⁶⁹. For instance, the prospect of a high distraction search could subtly influence participants’ sensory-mnemonic decisions towards using less memory, without necessitating any substantial strategic changes in encoding behaviour or interfered WM content. Further research is required to distinguish between potential drivers of distraction-induced changes to the coordination of WM usage and encoding.”

Additionally, we clarified the language in the paragraph discussing the overall low reliance on WM in *all* conditions (p 18, line 1):

“A related question is how to interpret the overall minimal reliance on WM ^{41,42} in all experimental conditions – even when distraction was low. We propose that relying on WM less can carry strategic adaptive advantages by dynamically reducing distractor interference (in line with ⁶⁷).[...]”

Further, we now report additional analyses testing for potential strategy differences on p. 14, line 1 and in *Supplementary Notes 3*:

“Importantly, the flexibility in self-structuring behaviour during the task did additionally allow for potential strategy changes. That is, behavioural sequences could have a different structure, even if participants are using the same number of attributes in WM. For example, using two attributes in WM could be the result of first searching for and picking up a target object (i.e., identity feature used) and then directly placing this object (i.e., location attribute used). At the same time, participants could have used two attributes in WM by first placing an already picked-up object (i.e., participant encoded while holding an object in hand) before continuing to search for and pick-up another object. However, while visual distraction affected WM usage, it did not lead to major strategy changes (see Supplementary Notes 3).”

2. Besides expected and exciting main result patterns, the authors found a couple unexpected results but did not address them sufficiently. Here are two examples. On page 4, the authors found an interaction between distraction and locomotive effort in model viewing time. Participants encoded slightly longer when distraction was high compared to low during the low locomotive effort condition. On page 7, there was an interaction between distraction and locomotive effort. Distraction decreased WM usage more when locomotive effort was high. Though some of those results might not be of key interest to the authors' main conclusion, it would be worthwhile to discuss and interpret this finding. For example, the observation that distraction decreased WM usage more when locomotive effort was high seems to suggest that distraction interference was larger in conditions where WM is more used than when it is not. This is additional evidence suggesting that changes in WM usage can be associated with distraction interference instead of strategy changes, which have been neglected across the text.

This is a great addition to the point raised above. As already mentioned, we now explicitly address the possibility of both distraction interference and potential strategy changes. Along this line, we also interpret the interaction mentioned by the reviewer (p. 18, line 6):

“In line with this proposal – and further strengthening the assumption that changes in WM usage can be associated with distraction interference instead of strategy changes – is the finding that distraction decreased WM usage more in conditions where WM is used more (i.e., high movement effort). [...]”

3. The finding that distractions primarily induced identity errors rather than location errors is intriguing, but I found it a bit confusing as to how these errors were separated. Identity errors would remain independent of location errors, considering that pickups occurred before placements. However, location errors could be contaminated by identity errors. When an incorrect item was picked up, the subsequent placement would also be inaccurate. I am curious whether the authors analyzed only trials in which correct items were picked up to assess location errors, or if they overlooked the distinction between correct and incorrect pickups. The latter scenario would imply that location errors might have been influenced by identity errors. It is necessary for the method section to explicitly address this concern, offering clarification on how the analysis was done.

We thank the reviewer for this important request for clarification. In addressing the concerns raised, we would like to clarify our approach to inclusions of errors.

As described by the reviewer, placement errors could indeed be contaminated by identity errors. Additionally, participants were required to correct any mistake before continuing with the task, causing a disruption which is not present in behavioural sequences without errors and potentially

affecting any actions subsequent to an error occurring. Therefore, errors were only included in the analysis if no other error (placement or pick-up) had occurred in the behavioural sequence prior to the error in question. That is, data for placement and pick-up errors were excluded after the first error occurred and subsequent errors did not enter data analysis in order to keep location (and identity) errors pure.

We now provide additional clarification regarding error analyses (p.10, line 6):

“Analyses of error rates were based on the same data used to analyse sensory-mnemonic decisions. [...] Given that incorrectly picked-up objects inadvertently led to a placement error if participants attempted to place a non-target object, combined with the general disruption caused by the need to correct errors, sequences were excluded after the first error occurred. Any subsequent errors within the same behavioural sequence therefore did not enter data analysis to keep location and identity errors pure.”

Minor points:

1. Bayes factors (BF) should be reported for all analyses.

We agree that this is an important addition to the presented analyses. We now report Bayes factors for all pairwise comparisons (see Supplementary information for full reporting on stats).

2. same for effect sizes

We now report all relevant effect sizes.

3. In fig 6, the y axis for all identity error figures should be aligned.

Great catch. We now aligned the y-axis in Figure 6.

4. In all figure legends, it should be specified what *, **, *** mean.

Great point! We've added the following sentence to all relevant figures:

*“The symbols *, **, and *** in the figure denote statistical significance with p-values less than 0.05, 0.01, and 0.001, respectively.”*

Reviewer #3 (Remarks to the Author):

The authors investigate the effects of visual distraction and locomotor effort on different components of an object copying task. They conclude that visual distraction interfered with encoding, visual search, working memory reliance, and decision-making. Higher locomotive effort also increased working memory reliance. I think the research question is interesting and fits nicely within the scope of the journal. The methodology is clearly explained, and the figures are nicely presented (I particularly appreciate the plotting of individual data points). However, I do have several queries regarding the authors' choice of terminology, and rationale for the experimental design and statistical analyses that I elaborate on below. I think these points need addressing in a revision.

We thank the reviewer for the positive evaluation of our work!

Major

1. I find the use of the term 'distraction' throughout the paper to be rather misleading and confusing. When I read the title, I expected the experiment to focus on 'distractions' in the sense of someone interrupting the naturalistic task currently being undertaken (e.g., the phone rings during the baking example that the authors give). However, it becomes apparent that the authors are referring to, and manipulate, characteristics of visual distractors. I think the narrative of the paper and terminology used could be adjusted to make this clearer for the reader.

Below are some examples:

- a. Results Section (Page 3, line 23), 'We first consider the consequences of distraction on summary measures...'. This sentence could be interpreted as 'distraction' referring to both the independent variables collectively (i.e., visual distractors and locomotor effort could both be distractions) or as referring to the effect of visual distraction variable, specifically.
- b. The sub-headings of the results section all highlight 'Distraction does...'. In line with my above comment, the use of this general term 'distraction' initially made me think the authors were referring to both variables collectively, but they are not. Rather, there is much more emphasis on the visual distraction effect compared with the locomotor effort effect.
- c. Discussion (Page 9, line 16). 'Using a novel VR method, we discovered dissociable and multifaceted consequences of distraction on several subcomponents of natural behaviour'. This implies 'distraction' is a multifaceted construct but in this experiment, it is specifically the opacity of a distractors.
- d. More generally, the authors swap between using 'visual distraction' and just 'distraction' to refer to the same thing. I think the paper would be clearer if consistent terminology was used throughout.

We understand the potential source for confusion and thank the reviewer for raising this important point. We fully agree that the overall narrative of the manuscript should be clearly framed in the context of **visual** distraction.

We therefore aimed to clarify the language throughout the manuscript. Most importantly, we amended the title of the manuscript which now reads "*Multifaceted consequences of **visual** distraction during natural behaviour*". We believe this will greatly decrease the ambiguity related

to the nature of the investigated form of distraction and prime readers correctly from the very beginning.

We also clarified our visual distraction manipulation in the Methods section (p.5, line 26):

“[...] Specifically, we changed the opacity of distractor objects in the resource pool (Fig 1 and Video 2), manipulating the discriminability between targets and distractors, and therefore distractibility, while participants searched for and picked up target objects. In the low-distraction condition, distractor objects were overlaid with a white plane set to 20% transparency. In the high-distraction condition, distractor objects appeared with the same opacity as target objects. We will henceforth denote this manipulation as distraction.”

2. The authors explain that the rationale for including locomotor effort as an independent variable was to ‘induce more variability in participants’ WM usage and investigate how the impact of distraction varies depending on locomotor demands (page 2, lines 47 – 50).’ From my understanding, the authors are interested in the interaction between visual distraction and locomotor effort. However, I miss the rationale explaining why there would be an interaction because they effect different subcomponents of the task. Specifically, visual distraction effects encoding and search whereas locomotion effects WM usage. I would appreciate some additional clarity for the rationale behind including locomotor effort as an independent variable, and the logic behind expecting an interaction.

3. Following on from the above, I would appreciate the rationale behind some of the statistical analyses performed and clarity on what hypotheses are being tested. From my understanding, the authors expect the two different independent variables to have an effect on different subcomponents of the task. Therefore, there are several analyses reported in the paper that I am not convinced are testing well-motivated hypotheses. For example, the effect of locomotive effort on display completion time (Page 4, lines 6-11) and the effect of locomotive effort on visual search time (Page 6, lines 2- 8).

We appreciate the feedback provided by the reviewer. We would like to clarify the general rationale behind including *movement* effort (previously *locomotive* effort) as an independent variable – in our study in general and, as an extension, in all statistical analyses.

The reviewer is right that the two manipulations were primarily intended to affect different subcomponents of the task. However, a key aim of the work presented here was to investigate whether and how the effect of visual distraction present in one subcomponent (i.e., visual search) would affect other key components of behaviour (i.e., WM usage combined with encoding).

Prior work has demonstrated extensively that reliance on memory during naturalistic task is generally low but increases when accessing information in the task environment becomes more effortful (e.g., Ballard et al., 1995; Draschkow et al., 2021; Hardiess et al., 2011; Somai et al., 2020). For example, as the distance between model and resources increases (i.e., requiring greater movement effort), participants spent more time encoding from the model as well as relying on WM more (Draschkow et al. 2021). That is, we can experimentally influence how much participants rely on WM (combined with how long participants encode) to guide behaviour.

At the same time, we expected that distraction present during visual search could act on other components of behaviour dependent on the overall WM usage. For example, when using a very minimal memory strategy in our task (i.e., using only one attribute in WM), participants can reduce distractor interference during target search. In this case, less information in WM needs to be protected from distraction.

Given this interlinked nature of potential effects of visual distraction and overall degree of reliance on WM, we included two movement effort conditions (both in our design and all statistical analyses). We now expand on this in our Method section (p. 5, line 33), highlighting the aim of capturing potential differences in the effect of distraction in response to overall reliance on WM:

“Prior work demonstrated that reliance on memory during naturalistic tasks is generally low (Draschkow et al., 2021; Droll & Hayhoe, 2007) but increases when accessing information in the task environment becomes more effortful (e.g., Ballard et al., 1995; Draschkow et al., 2021; Hardiess et al., 2011; Somai et al., 2020). For example, as the distance to the model increases (i.e., requiring greater movement effort to look back to the model), participants spend more time encoding from the model as well as relying on WM more (Draschkow et al. 2021). Here, we manipulated movement effort to experimentally induce more variability in how much participants relied on WM, allowing us to observe the effects of visual distraction during a broader range of naturalistic memory usage.”

4. Results: ‘Distraction reduces memory usage’. My interpretation is that the most compelling effect in Figure 5b is that of locomotor effort on memory usage, yet the authors don’t really comment on this. Given the focus of the paper on naturalistic tasks, it seems more relevant that rotating 90 degrees has such a clear effect on working memory usage. In my opinion, the effect of distraction is less compelling, especially when you consider that only integer values are possible for the number of attributes measure (although there is consistency in the downward slope of the participant level data, these are mostly within an single attribute value). I wonder whether the authors think that participants are doing something meaningfully different in the high compared with low distraction condition or are they mostly using the same number of attributes? I do think this is a nice measure and would be interested to see the distribution of attributes in WM, which would help address my query above.

Thank you for this important point. First, we would like to clarify that we focused on *movement effort* in a prior publication (see Draschkow et al., 2021). However, while the focus of our study is visual distraction, we agree that the effect of distraction could be interpreted more comprehensively in relation to the movement effort effects (also see response to reviewer #1, major comment 3).

To address this, as the reviewer suggests, we added an analysis of the probability of using different numbers of attributes in memory to the manuscript and show these results in Figure 5C (p. 13, line 37):

“Comparing the probability of using different numbers of attributes in WM (Fig. 5C) further highlights that the distraction-induced difference in WM usage was small and consistent (see Supplementary information, Table 9 for a full reporting of the pairwise comparisons).”

Figure 5. Trade-off between reliance on WM and gathering information from the external world. a) Our implicit metric for working memory (WM) usage: Copying each object requires its identity and location information (attribute) to be held in memory. Counting successful pick-ups (i.e., identity attribute used) and placements (i.e., location attribute used) in between model fixations provided a metric for the number of attributes used in WM. **b)** Average number of attributes used in WM in both movement effort conditions as a function of distraction, **c)** Distribution of attributes used in WM (%). Error bars depict the standard error of the mean. Lines show individual participant data. N shows the number of overall sequences included in the analysis. The symbols *, **, and *** in the figure denote statistical significance with p-values less than 0.05, 0.01, and 0.001, respectively.

5. The section on sensory-mnemonic decisions is interesting and highlights the sequential nature of this task. I think an important consideration is how the participants' behaviour changes over the course of the task because of the task itself (i.e., rather than because of the visual distraction). My understanding is that all aspects of the task become easier as the participant moves more blocks from the resource pool to the workspace because there are less items to search in the resource pool, and less possible locations to place the item in the workspace. I wonder whether the authors have thought about the implications of this for their results. For example, I would expect the attributes in WM measure to increase over time (less reliance on the model) because there is less need to verify the correct location for an item once the workspace becomes fuller (i.e., knowing an item is roughly top left becomes sufficient rather than needing to know the position is [1,2] in the array). I think a visualisation showing how some of the measures change over the course of the task would benefit the manuscript because this seems an important consideration in the task.

We thank the reviewer for raising this important point. We fully agree that temporal changes over the course of one trial is an interesting and worthwhile consideration; warranting a careful investigation, which we are currently completing in a separate set of experiments.

First, we would like to clarify that placed objects reappeared on the resource pool (see p. 5, line 8), keeping the number of items in the resource pool constant over the course of the trial. Nevertheless, there are several factors that potentially change over the course of a trial: Already placed objects can serve as anchors/cues while placing the next object (as also noted by the reviewer), demands on monitoring and planning (i.e., knowing which objects have been placed already and which object to copy next) likely differ at the beginning vs middle vs end of a trial, and constraints on some of the metrics shift (e.g., the maximal number of attributes one can use in memory decreases over the course of the trial).

To investigate potential temporal effects in the present data set, we included trial progress (i.e., number of targets already placed at given point in trial) as an additive predictor to the final models of metrics in different subcomponents of behaviour (i.e., search time, model viewing time and attributes used in memory). As expected by the reviewer, trial progress did indeed affect behaviour across multiple subcomponents, underscoring that temporal effects are an important dimension in extended tasks. However – and critical for the current manuscript –, including trial progress as a predictor did not affect our conclusions regarding the effects of visual distraction, movement effort, or their interaction.

We now present these results in *Supplementary Note 2* and highlight that the effects of distraction and movement effort go above and beyond any temporal task effects for each subcomponent (i.e., Encoding, Search and WM usage). For example, on p. 11, line 38:

“Given the extended nature of each trial, we additionally investigated potential temporal effects in the encoding subcomponent, showing that the effects of distraction and movement effort go above and beyond any temporal task effects (see Supplementary Notes 2).”

Supplementary Note 3, (p.2, line 22 in the Supplementary Information):

*“Results. Specifically, trial progress predicted model viewing time, including both a linear ($\beta = -3.50$, $SE = 0.36$, $t = -9.84$, $p < 0.001$, $CI_{95\%} = [-4.19, -2.80]$) and quadratic ($\beta = -4.83$, $SE = 0.35$, $t = -13.63$, $p < 0.001$, $CI_{95\%} = [-5.53, -4.14]$) effect (see **Fig. S2A**). However, even after the inclusion of trial progress as a predictor, we again observed no meaningful difference in model viewing times between distraction conditions ($\beta = 0.01$, $SE = 0.03$, $t = 1.88$, $p = 0.07$, $CI_{95\%} = [-0.0003, 0.01]$). Additionally, movement effort still predicted model viewing time ($\beta = -0.19$, $SE = 0.02$, $t = -11.43$, $p < 0.001$, $CI_{95\%} = [-0.22, -0.15]$). That is, participants still encoded from the model for longer when movement effort was high. Further, the effect of distraction on model viewing time was weakly but reliably moderated by movement effort ($\beta = 0.01$, $SE = 0.003$, $t = 2.39$, $p = 0.02$, $CI_{95\%} = [0.001, 0.01]$).*

*Trial progress also predicted attributes used in WM, including both a linear ($\beta = -16.31$, $SE = 0.71$, $z = -22.89$, $p < 0.001$, $CI_{95\%} = [-17.70, -14.91]$) and quadratic ($\beta = -3.09$, $SE = 0.64$, $t = -4.85$, $p < 0.001$, $CI_{95\%} = [-4.33, -1.84]$) effect (see **Fig. S2B**). After including trial progress as a predictor, high distraction again decreased ($\beta = -0.03$, $SE = 0.004$, $z = -7.04$, $p < 0.001$, $CI_{95\%} = [-0.03, -0.019]$) and high movement effort increased ($\beta = -0.21$, $SE = 0.01$, $z = -17.99$, $p < 0.001$, $CI_{95\%} = [-0.23, -0.18]$) the number of attributes used in WM. Distraction also interacted with movement effort ($\beta = 0.01$, $SE = 0.003$, $z = 2.32$, $p = 0.02$, $CI_{95\%} = [0.001, 0.01]$).*

We additionally found a linear ($\beta = -2.79$, $SE = 0.43$, $t = -6.43$, $p < 0.001$, $CI_{95\%} = [-3.64, -1.93]$) and quadratic ($\beta = 2.15$, $SE = 0.43$, $t = 4.95$, $p < 0.001$, $CI_{95\%} = [-0.23, -0.18]$) effect of trial progress on search time (see Fig. S2C). After including trial progress as a predictor, high distraction still slowed down visual search ($\beta = 0.1$, $SE = 0.004$, $t = 26.15$, $p < 0.001$, $CI_{95\%} = [0.09, 0.11]$) but we observe no difference in search time between movement effort conditions, $\beta < 0.001$, $SE = 0.004$, $t = 0.11$, $p = 0.92$, $CI_{95\%} = [-0.01, 0.01]$). Distraction and movement effort also again interacted ($\beta = 0.01$, $SE = 0.003$, $t = 2.32$, $p = 0.03$, $CI_{95\%} = [0.001, 0.01]$). “

Figure S2. Temporal task effects on metrics from different subcomponents. Trial progress (as indexed by the number of target objects already placed at a given time within the trial) affected metrics across multiple subcomponents (i.e., **a**) Encoding as indexed by model viewing times, **b**) WM usage as indexed by Attributes used in WM, and **c**) Visual search as indexed by search time). Error bars depict standard error of the mean ($N = 30$).

6. Overall, I find the discussion rather difficult to follow. The authors make claims about ‘dissociable’, ‘multifaceted’, ‘cascading’ consequences of distraction but I struggle to understand what they really mean by these terms. Specifically:

a. What is multifaceted? The ‘distraction’ investigated here is very specific (i.e., visual distraction manipulated by changing opacity).

b. What consequences of distraction are dissociated? Do the authors mean there is a different consequence of distraction in the different parts of the task.

c. What does cascading refer to? This is almost contradictory to ‘dissociating’. It implies that the effect of visual distraction in one component of the task has an effect on another.

I would appreciate the authors providing a clearer narrative throughout the discussion and suggest considering using different terminology.

We have substantially revised the discussion section, with the aim of enhancing its clarity and coherence. We hope that these revisions have made it easier to follow and understand.

Minor

1. Abstract. 'Interfered with' is rather vague. Can this be more specific by referring to the DV (e.g., a decrease in WM load).

We now directly specify how distraction affected the DV and only use "interference" as a general term when discussing overall result patterns.

2. 'HDM' should be 'HMD' in a few places (e.g., page 11, line 36)

Thanks for spotting this mistake! We've corrected all abbreviations of HMD.

3. Figure 1a. In this figure, the resource pool is 3 rows*8 columns. It would be nice if this were consistent with the rest of the manuscript (both figures and text).

We apologise for any confusion. We believe the reviewer is referring to the fact that the resource pool is depicted from a birds-eye perspective in Fig. 1A (appearing as 8 rows* 3 columns), but from the perspective of standing in front of it as a participant in all other Figures (3 rows* 8 columns).

We believe that rotating all of Fig 1A to align the orientation of the resource pool with the other figures would greatly decrease the overall readability of this figure panel, but now clarify in the text that "3 rows of 8 cubes" refers to the perspective of standing in front of the resource pool (p.4, line 37):

"Within the resource pool, participants would find an arrangement of 24 cubes (10 x 10 cm), organized in three rows of 8 cubes, as viewed from a frontal perspective, [...]."

4. Page 12, line 28. 'Participants could not pick up another object until their mistake was corrected.' To what extent does this part of the experiment constrain the decision process and what are the implications of this for your results?

This is a good question, and we agree that the rationale and implications of this constraint should be made explicit. We included this constraint for two reasons: First, participants sometimes did not realise they made a mistake (e.g., they rotated away before seeing that the object was placed incorrectly and would have picked up another object). We prohibited them from picking up another object to ensure that any placement error was corrected before moving on with the next task action to minimise the impact of errors on subsequent behaviour. Secondly, it enforced a strict sequential nature to the task, allowing us to better align behavioural sequences. For example, without this constraint, participants could first move *multiple* target objects from the resource to the workspace by placing them on random (most likely incorrect) slots, before placing them all in their correct slots.

We have added the following clarification to the "Procedure and task" section (p. 5, line 12):

“Participants could not pick up another object until the previously picked-up object was placed correctly: all other objects in the resource pool disappeared until the not-yet (correctly) placed object was placed correctly or brought back to the resource pool. This ensured that a) participants corrected their mistake before continuing with the task, preventing any flow-over of mistakes into the next task action and b) completed the task in a sequential object-by-object manner, prohibiting them from moving multiple target objects from the resource pool to the workspace before placing them in their corresponding placeholders..”

5. Frame durations. What does this mean? Does each frame not have the same duration (90 Hz)?

Recorded frame durations are linked to the length of time between Update function calls in Unity, which is not necessarily always constant (as with most software applications). For example, momentary changes in CPU load could result in slight changes in time between updates. Frame duration therefore refers to the actual time difference between recorded frame updates (with an average and target sampling rate of 90 Hz).

6. I think ‘locomotive effort’ could be better replaced with ‘locomotor effort’ throughout the manuscript because it’s the participant that moves.

Thanks for pointing this out. We originally named our secondary experimental manipulation “locomotive effort” to be consistent with our previous work. We agree with all reviewers that this could lead to some ambiguous interpretations. Because the experimental manipulation made participants *move* their entire body, we have now renamed it “movement effort”.

18th Apr 24

Dear Mx Kumle,

Your manuscript titled "Multifaceted consequences of visual distraction during natural behaviour" has now been seen by our reviewers, whose comments appear below. In light of their advice I am delighted to say that we are happy, in principle, to publish a suitably revised version in *Communications Psychology* under the open access CC BY license (Creative Commons Attribution v4.0 International License).

As you will see from their reports, especially Reviewer #3 maintains several hesitations; we consider this important feedback. Your final submission needs to account for these concerns, in particular through revisions to the Discussion and Limitations sections.

We therefore invite you to revise your paper one last time to address the remaining concerns of our reviewers and a list of editorial requests. At the same time we ask that you edit your manuscript to comply with our format requirements and to maximise the accessibility and therefore the impact of your work.

EDITORIAL REQUESTS:

SUBMISSION INFORMATION:

OPEN ACCESS:

Communications Psychology is a fully open access journal. Articles are made freely accessible on publication under a CC BY license (Creative Commons Attribution 4.0 International License). This license allows maximum dissemination and re-use of open access materials and is preferred by many research funding bodies.

For further information about article processing charges, open access funding, and advice and support from Nature Research, please visit <https://www.nature.com/commspsychol/article-processing-charges>

At acceptance, you will be provided with instructions for completing this CC BY license on behalf of all authors. This grants us the necessary permissions to publish your paper. Additionally, you will be asked to declare that all required third party permissions have been obtained, and to provide billing information in order to pay the article-processing charge (APC).

* **DATA AVAILABILITY:**

[link redacted]

Best regards,

Marike

Marike Schiffer, PhD

Chief Editor

Communications Psychology

REVIEWERS' COMMENTS:

Reviewer #2 (Remarks to the Author):

All my previous comments have been well addressed by the authors. I have only one minor suggestion to make. I appreciate the author's proposal that "relying on working memory less can have strategic adaptive advantages by dynamically reducing distractor interference." This explanation could potentially be linked to the prioritization effect in working memory and its role in protecting against distractions. It might be worth discussing this aspect in the final discussion section.

Reviewer #3 (Remarks to the Author):

I would like to thank the authors for their consideration of the points that I raised and the substantial revisions. I think the paper nicely introduces a novel VR approach for evaluating naturalistic behaviour to this research area. However, I find the narrative of the paper is rather disjointed and biased towards reporting the effect of visual distraction despite the experimental design and statistics presented. I think there are two negative consequences of this: 1) sometimes overstating the effect of visual distraction and 2) limited interpretation of the effect of movement effort. The lack of focus on the latter makes it seem that not all aspects of the naturalistic behaviour being studied are considered.

Major Comments:

1. I thank the authors for clarifying the reason for including movement effort as an independent variable. Nevertheless, I still find the narrative of the paper rather disjointed. Despite the 2-x-2 design that is used in all the statistics and figures, there seems to be a consistent bias to focus on the effect of visual distraction in the text. For example, there is no mention of movement effort in the abstract, Figure 1b, or results subheadings. This contributed to my previous comment that there are a lot of statistics presented in this paper that do not seem to be well-motivated tests of hypotheses (see Brenner, 2016). I think the narrative of the paper can be improved to better link the rationale of the experiment to the experimental design and results. Specifically, it is not clear to me why the authors focus the paper so heavily on visual distraction.

Brenner, E. (2016). Why we need to do fewer statistical tests. *Perception*, 45(5), 489-491.

2. I appreciate the addition of Figure 5c showing the distribution of attributes in WM. However, I do not follow the authors' reasoning for concluding that 'distraction-induced difference in WM usage was small and consistent' (Page 13, line 38). Visualisation of Figure 5c and the t-values in Table 9 (supplementary material) show that the direction of the effect of visual distraction changes, dependent upon the number of attributes in WM. I still think the most interesting finding is the effect of movement effort on attributes in WM; the distributions are clearly different for the low and high movement effort conditions. In line with my previous comment, I therefore struggle to understand the authors' focus on visual distraction. The subheading reads 'Distraction decreases memory usage' but I am still not convinced there is a meaningful effect of visual distraction on the number of attributes in WM. In contrast, there is a very clear effect of movement effort.

3. Following from my comment above, I'm not convinced that the authors are really capturing the number of attributes in WM. Based on my understanding of Figure 5a, the measure captures how many 'task steps' (i.e., pick up and places) a participant does before returning to fixate the model. It thus seems possible that participants hold more attributes in WM but, for a variety of reasons (e.g., forgetting, confidence, ease), choose to refer back to the model. For example, the reason participants primarily only hold one attribute in WM according to this measure when in the 0-degree condition might be because there is no real cost to them quickly checking the model which is directly in front of them as they place the object. This interpretation again relies on the consideration of movement effort. I think the discussion would benefit from consideration of what factors influence a participant's decision to use WM, which is what I think this measure is also capturing.

4. I find the results on sensory mnemonic decisions surprising. My understanding is that participants are less likely to pick up a second object but more likely to place a second object in the workspace in the high visual distraction condition. Does this imply that the reliance on memory depends upon whether it is location or identity information being used, with participants being more confident in their location memory in the high distraction condition? Do the authors have an explanation for this finding? I think this could be explored more in the discussion.

5. Discussion. I find the structure of this section very different to the generally well-structured paper which takes the reader through each sub-component sequentially (i.e., encoding, search, WM usage, decision-making). Here, it seems there is a bias to focus on WM usage and there remains little interpretation of the results relating to movement effort. Some more specific points:

a. The first paragraph reports 'We show that visual distractionlead to a subtle change in subsequent sensory mnemonic decisions...' (Page 17, line 3) which is immediately very focussed on the sensory mnemonics decisions subcomponent specifically rather than addressing the broader aim in the introduction to 'investigate the influences of visual distraction on several cognitive subcomponents of naturally unfolding behaviour.'

b. 'Memory-guided natural behaviour' – the term is used quite frequently in the discussion despite only being mentioned once before and not being clearly defined.

c. Focal vs cascading vs cumulative? These terms all seem rather contradictory to me. 'Focal' implies distraction effects one specific subcomponent, 'cascading' implies it effects all subcomponents sequentially, 'cumulative' implies the effect is additive. What results are the authors using as evidence for focal, cascading, and cumulative effects?

d. What do the authors mean by multifaceted? The 'distraction' investigated here is very specific (i.e., visual distraction).

Minor Comments:

1. I appreciate the authors clarifying that they are referring to visual distraction in the title and abstract. Nevertheless, I don't think this has been incorporated consistently throughout the remainder of the paper (e.g., the sub-headings of the results section). I think this remains an important point to address because sentences such as 'Effects of distraction on core components of natural behaviour' (Page 2, line 10) strongly imply to me that the paper will investigate common distractions in naturalistic behaviour (e.g., phone ringing, cup falling on the floor) rather than the rather artificial distraction of item opacity.

2. Thank you for highlighting that the items in the resource pool reappeared after being moved and for presenting the data looking at temporal effects in Supplementary Notes 2. I proposed this as an exploratory analysis and think it is sufficient to just include the figures to allow a visualisation of temporal effects rather than statistics. Given the focus of the paper on naturalistic behaviour, I think the main text would benefit from the inclusion of this analysis which clearly shows temporal effects (although I do acknowledge this may not be possible due to word limits).

3. The conclusions from the section 'Distraction decreases memory usage' do not really fit with the introduction to the section 'Consequences of distraction on sensory-mnemonic decisions.' I acknowledge the authors addition of the word 'slightly' but do not agree that Figure 5 supports the conclusion 'When encountering high distraction, participants use a slightly lower number of attributes in WM' (Page 14, line 23). The integer number is the same. The authors acknowledge this themselves 'participants predominantly relied on only one attribute when movement effort was low' (Page 15, line 11). Here it becomes clear why the movement effort variable was included.

4. I find some statements are rather vague and would benefit from more specific terminology. For example, 'how specific perturbations in sensory parameters of the environment can have multiple cascading consequences for processes that are contingent on the affected subcomponent.' (Page 3, line 16). 'Perturbations in sensory parameters of the environment' sounds very elaborate given the actual manipulation is the opacity of the items.

5. Thank you for the explanation on Figure 1a. Perhaps it's helpful to rotate the heading 'resources' to be in the participants viewpoint as well and be consistent with the heading 'model'.

REVIEWERS' COMMENTS:

Reviewer #2 (Remarks to the Author):

All my previous comments have been well addressed by the authors. I have only one minor suggestion to make. I appreciate the author's proposal that "relying on working memory less can have strategic adaptive advantages by dynamically reducing distractor interference." This explanation could potentially be linked to the prioritization effect in working memory and its role in protecting against distractions. It might be worth discussing this aspect in the final discussion section.

We thank the reviewer for the positive feedback. We agree that linking our proposal with potential WM prioritisation effects would improve our Discussion. We now discuss this accordingly on page 14:

"We propose that relying on WM less can carry strategic adaptive advantages by dynamically reducing distractor interference (in line with Hamblin-Frohman & Becker, 2023). For example, when using a minimal memory strategy in our task, less information in WM needs to be selected for prioritisation and protected from external sources of distraction (Allen & Ueno, 2018; Lorenc et al., 2021; Makovski & Jiang, 2007; Makovski & Pertzov, 2015; Zhang & Lewis-Peacock, 2022). Additionally, keeping the number of items in WM at a minimum could reduce internal inter-item competition between items held in WM concurrently (Bouchacourt & Buschman, 2019; Czoschke et al., 2023; Pertzov et al., 2017)."

Reviewer #3 (Remarks to the Author):

I would like to thank the authors for their consideration of the points that I raised and the substantial revisions. I think the paper nicely introduces a novel VR approach for evaluating naturalistic behaviour to this research area. However, I find the narrative of the paper is rather disjointed and biased towards reporting the effect of visual distraction despite the experimental design and statistics presented. I think there are two negative consequences of this: 1) sometimes overstating the effect of visual distraction and 2) limited interpretation of the effect of movement effort. The lack of focus on the latter makes it seem that not all aspects of the naturalistic behaviour being studied are considered.

Major Comments:

1. I thank the authors for clarifying the reason for including movement effort as an independent variable. Nevertheless, I still find the narrative of the paper rather disjointed. Despite the 2-x-2 design that is used in all the statistics and figures, there seems to be a consistent bias to focus on the effect of visual distraction in the text. For example, there is no mention of movement effort in the abstract, Figure 1b, or results subheadings. This contributed to my previous comment that there are a lot of statistics presented in this paper that do not seem to be well-motivated tests of hypotheses (see Brenner, 2016). I think the narrative of the paper can be improved to better link the rationale of the

experiment to the experimental design and results. Specifically, it is not clear to me why the authors focus the paper so heavily on visual distraction.

Brenner, E. (2016). Why we need to do fewer statistical tests. *Perception*, 45(5), 489-491.

Thank you for appreciating the novel contribution of the paper in introducing a new framework for evaluating immersive behaviour and for elaborating on the concerns regarding our narrative. Like the reviewer, we are strongly committed to promoting and championing robust and reproducible (as well as open) science.

Specifically in relation to the pattern of findings grounding the current investigation, we feel we are building on a robust experimental foundation. While research on naturalistic WM usage and its coordination with encoding from the external environment is still sparse, it is crucial to highlight the consistent and well-established finding that this coordination is intricately linked to the effort required for encoding (see Draschkow et al., 2021; Droll & Hayhoe, 2007; Gray et al., 2006; Hardiess et al., 2011; Somai et al., 2020). That is, WM usage is generally low but increases with increased effort related to encoding. In this context, our previous work has already focussed in detail on the effect of *movement* effort (see Draschkow et al., 2021).

An important and fundamental take-away from this line of research is that WM usage and its coordination with encoding from the environment in naturalistic settings is not fixed but variable. We believe that capturing this variability is important when considering the influences of other effects – like visual distraction – on WM usage and encoding. Specifically, factors such as distraction could plausibly have different effects on WM usage when participants are relying more on encoding from the external environment (i.e., WM use is low) vs. more on information in WM (i.e., WM use is high/increased). Indeed, we found that the effect of distraction was more pronounced when WM use was higher.

Informed by the consistent findings on the effect of effort on WM usage, we therefore used movement effort as a *tool* to experimentally induce variability in WM usage. We describe the inclusion of movement effort as such within the manuscript, in both the Introduction (p. 2):

“Working memory usage is generally low during natural behaviour⁴¹⁻⁴⁵ but increases when additional movement effort is required to access information in the external environment⁴². To experimentally induce more variability in participant’s WM usage and investigate how the impact of distraction varies depending on locomotor demands, we therefore also included two movement effort conditions (Fig1A, Supplementary Movie 3).”

and Methods (p.5):

“Second, the model’s location was varied (0° or 90° relative to the workspace, Fig 1A and Supplementary Movie 3), thus manipulating the movement effort associated with encoding information from the model (Draschkow et al., 2021). Prior work demonstrated that reliance on memory during naturalistic tasks is generally low (Draschkow et al., 2021; Droll & Hayhoe, 2007, 2007) but increases when accessing information in the task environment becomes more effortful

(Draschkow et al., 2021; Hardiess et al., 2011; Somai et al., 2020). For example, as the distance to the model increases (i.e., requiring greater movement effort to look back to the model), participants spend more time encoding from the model as well as relying on WM more (Draschkow et al., 2021). Replicating previous work (Draschkow et al., 2021), we manipulated movement effort to experimentally induce more variability in how much participants relied on WM, allowing us to observe the effects of visual distraction during a broader range of naturalistic memory usage.”

Additionally, to add further context for the results on movement effort, we now more clearly highlight that the findings replicate previous work by Draschkow et al. (2021) in the introduction (p. 3):

“To foreshadow, this study makes three distinct contributions. First, we replicate that reliance on WM increased as sampling information from the environment required increased locomotion (Draschkow et al., 2021). Second, we develop analytical tools that segment continuous behaviour into its core subcomponents. Third, we illustrate how specific changes in sensory parameters of the environment can have multiple cascading consequences for processes that are contingent on the affected subcomponent.”

and results (p. 12):

“As intended, the average number of attributes used in WM increased with movement effort ($\beta = -0.21$, $SE = 0.01$, $z = -18.17$, $p < 0.001$, $CI_{95\%} = [-0.23, -0.18]$)^{42,44}, allowing us to observe a broader range of natural WM usage (Fig. 5B). Further replicating findings from Draschkow et al. (2021), participants mostly relied on only one attribute in WM when locomotive effort was low but were more likely to rely on 2 attributes in WM when movement effort was high (Fig. 5C).”

The focus on visual distraction therefore reflects the unique contributions of the manuscript to the literature on naturalistic memory-guided behaviour. We hope we could now clarify the intention with which movement effort was included in the experimental design, both in the response and in the manuscript.

2. I appreciate the addition of Figure 5c showing the distribution of attributes in WM. However, I do not follow the authors' reasoning for concluding that 'distraction-induced difference in WM usage was small and consistent' (Page 13, line 38). Visualisation of Figure 5c and the t-values in Table 9 (supplementary material) show that the direction of the effect of visual distraction changes, dependent upon the number of attributes in WM. I still think the most interesting finding is the effect of movement effort on attributes in WM; the distributions are clearly different for the low and high movement effort conditions. In line with my previous comment, I therefore struggle to understand the authors' focus on visual distraction. The subheading reads 'Distraction decreases memory usage' but I am still not convinced there is a meaningful effect of visual distraction on the number of attributes in WM. In contrast, there is a very clear effect of movement effort.

We thank the reviewer for pointing out the ambiguity in the sentence on page 13. We have now clarified this:

“Comparing the probability of using different numbers of attributes in WM (Fig. 5C) further highlighted that the distraction-induced difference in WM usage was consistently driven by a small decrease in using 3 or 4 attributes (see Supplementary Table 9 for a full reporting of the pairwise comparisons).”

We now also comment on the clear shift in the probabilities of using different numbers of attributes in WM between movement effort conditions; highlighting that these replicate findings reported in Draschkow et al. (2021) (p. 11):

“As intended, the average number of attributes used in WM increased with movement effort ($\beta = -0.21$, $SE = 0.01$, $z = -18.17$, $p < 0.001$, $CI_{95\%} = [-0.23, -0.18]$) (Draschkow et al., 2021; Hardiess et al., 2011), allowing us to observe a broader range of natural WM usage (Fig. 5B). Further replicating findings from Draschkow et al. (Draschkow et al., 2021), participants mostly relied on only one attribute in WM when locomotive effort was low, but were more likely to rely on 2 attributes in WM when movement effort was high (Fig. 5C).[...]”

3. Following from my comment above, I'm not convinced that the authors are really capturing the number of attributes in WM. Based on my understanding of Figure 5a, the measure captures how many 'task steps' (i.e., pick up and places) a participant does before returning to fixate the model. It thus seems possible that participants hold more attributes in WM but, for a variety of reasons (e.g., forgetting, confidence, ease), choose to refer back to the model. For example, the reason participants primarily only hold one attribute in WM according to this measure when in the 0-degree condition might be because there is no real cost to them quickly checking the model which is directly in front of them as they place the object. This interpretation again relies on the consideration of movement effort. I think the discussion would benefit from consideration of what factors influence a participant's decision to use WM, which is what I think this measure is also capturing.

The reviewer is correct in drawing a distinction between the number of attributes available in WM vs the number of attributes that the participant uses. We have also emphasised this important distinction in this and in our previous study (Draschkow et al. 2021). Using behavioural measures in our task, it is only possible to measure the number of attributes that are used in WM, which in principle can contain more content. We thus refer to this variable as “Attributes used in WM” throughout the manuscript (and previous work) and refrain from any interpretation related to content in WM.

We further fully agree that the factors that might influence the decision to use WM and are worth discussing. Hence, we discuss important distinctions between WM use and content on page 14:

“Our results suggest that visual distraction can also compromise WM usage during natural behaviour, prompting participants to encode from the external environment more frequently. In our task, however, WM usage could have been impacted by both distraction interference (i.e., more frequent encoding served to compensate for such interference) and/or proactive shifts in the coordination of encoding and WM usage. A strong example for a proactive shift in

WM usage can be found in participant's response to increased movement effort, prompting participants both to encode more objects and subsequently decrease their encoding frequency through an increased reliance on WM (see Draschkow et al., 2021) for a more detailed discussion). In contrast, the change in WM usage and encoding in response to distraction seems to have been reactive. Specifically, distraction did not lead to a systematic change in *how* participants encoded from the environment (i.e., the number of objects encoded, or time spent in each encoding episode) but only *how often* participants referred to the model – particularly when behaviour must be guided by information closely tied to the present distraction. In line with existing research on reactive distraction interference in WM (Liesefeld et al., 2020; Lorenc et al., 2021), an increase in encoding frequency could serve to compensate for impaired content in WM. Alternatively, while using WM necessitates the appropriate content to be present, additional factors may influence the decision to act on it (Dunn & Risko, 2016; Risko & Gilbert, 2016; Sahakian et al., 2023). For instance, the prospect of a high distraction search could influence participants' sensorimnemonic decisions towards using less memory, without necessitating any substantial strategic changes in encoding behaviour or impaired WM content. Further research is required to distinguish between potential drivers of distraction-induced changes in the balance of WM encoding and usage."

4. I find the results on sensory mnemonic decisions surprising. My understanding is that participants are less likely to pick up a second object but more likely to place a second object in the workspace in the high visual distraction condition. Does this imply that the reliance on memory depends upon whether it is location or identity information being used, with participants being more confident in their location memory in the high distraction condition? Do the authors have an explanation for this finding? I think this could be explored more in the discussion.

We agree that this pattern of consequences on sensorimnemonic decisions is intriguing and have also realised that the nature of the attribute may play an important role. As highlighted in our limitations section: which sensorimnemonic decisions are affected could be linked to the source of distraction. In the current study, we manipulated the perceptual visual similarity between irrelevant objects and the encoded identity information but did not manipulate the similarity of location information. At the same time, we find evidence for interference with identity-related decisions but not location-related decisions. However, more investigations of different types of distraction are needed to corroborate this explanation.

We revised the limitations section, relating it more clearly to the findings on sensorimnemonic decisions (p. 15):

"[...] Importantly, we do not wish to claim that the particular pattern of consequences we observed holds for all forms of distraction. For instance, we only find evidence for distraction interference with identity-related sensorimnemonic decisions. Crucially, in the work reported here, we introduced a specific source of visual distraction: we increased the perceptual visual similarity between irrelevant objects and the encoded identity information – known to contribute to the effect of distraction – but did not manipulate the similarity of location information. Distraction during natural behaviour, however, can take various forms: [...] Our study focused on one particularly powerful form of distractor interference, and future research is necessary to uncover consequences of other types of distraction."

5. Discussion. I find the structure of this section very different to the generally well-structured paper which takes the reader through each sub-component sequentially (i.e., encoding, search, WM usage, decision-making). Here, it seems there is a bias to focus on WM usage and there remains little interpretation of the results relating to movement effort. Some more specific points:

The focus on visual distraction reflects the unique contributions of the manuscript to the literature on naturalistic memory-guided behaviour. We hope that our discussion of the interaction between movement effort and distraction conditions highlights the importance of movement effort as a tool to induce more variability in WM usage (p.14):

“A related question is how to interpret the overall minimal reliance on WM ^{41,42} across all experimental conditions – even when distraction was low. We propose that relying on WM less can carry strategic adaptive advantages by dynamically reducing distractor interference (in line with ⁷¹). [...] In line with this proposal – and further strengthening the assumption that changes in WM usage can be associated with distraction interference instead of strategy changes – is the finding that distraction decreased WM usage more in conditions where WM is used more (i.e., high movement effort).”

Below, we respond to individual points made by the reviewer.

a. The first paragraph reports ‘We show that visual distractionlead to a subtle change in subsequent sensory mnemonic decisions...’ (Page 17, line 3) which is immediately very focussed on the sensory mnemonics decisions subcomponent specifically rather than addressing the broader aim in the introduction to ‘investigate the influences of visual distraction on several cognitive subcomponents of naturally unfolding behaviour.’

We hope we can clarify that we aimed to use the first two paragraphs of the discussion to synthesise the experimental effects across all analysed subcomponents of naturally unfolding behaviour: encoding, WM usage, and sensorimnemonic decisions. The aim of the follow-up paragraphs was to contextualise the most important and novel findings from our rich set of data, with a focus on visual distraction.

b. ‘Memory-guided natural behaviour’ – the term is used quite frequently in the discussion despite only being mentioned once before and not being clearly defined.

We thank the reviewer for highlighting the lack of a clear operationalisation of “memory-guided natural behaviour”. We now clarify our operationalisation within the introduction (p.2):

“Compared to traditional laboratory studies, participants could freely decide when to look back at or move away from the model display (i.e., stop and start encoding). That is, participants could choose between using their memory representations to guide behaviour (memory-guided behaviour) and looking back to objects within the model.”

c. Focal vs cascading vs cumulative? These terms all seem rather contradictory to me. 'Focal' implies distraction effects one specific subcomponent, 'cascading' implies it effects all subcomponents sequentially, 'cumulative' implies the effect is additive. What results are the authors using as evidence for focal, cascading, and cumulative effects?

We thank the reviewer for the important feedback. The reviewer's interpretation of the terms *focal*, *cumulative*, and *cascading* aligns with our intentions. We do not see these terms as complementary rather than contradictory: focal refers to effects originating within a specific stage or processing; cascading refers to the consequences that the original effect has at other, downstream stages; cumulative refers to the fact that the effects at various stages add up to impact the final behavioural outputs.

To add further clarification, "focal" is referring to the fact that distraction was introduced at a specific, isolated stage within the extended task. We now clarify this in the discussion:

Discussion, p. 14: "Overall, our results illustrate that focal effects of visual distraction (i.e., distraction was introduced at a specific, isolated stage within the extended task) on memory-guided natural behaviour can have downstream, cascading consequences on processes that are contingent on the distraction-affected event."

d. What do the authors mean by multifaceted? The 'distraction' investigated here is very specific (i.e., visual distraction).

As per our comment above, we wish to highlight that we only refer to "multifaceted" when discussing the *consequences* of distraction. For example, on p.13:

"Using a multivariate VR approach, we discovered multifaceted consequences of visual distraction on several interconnected subcomponents of natural behaviour."

Minor Comments:

1. I appreciate the authors clarifying that they are referring to visual distraction in the title and abstract. Nevertheless, I don't think this has been incorporated consistently throughout the remainder of the paper (e.g., the sub-headings of the results section). I think this remains an important point to address because sentences such as 'Effects of distraction on core components of natural behaviour' (Page 2, line 10) strongly imply to me that the paper will investigate common distractions in naturalistic behaviour (e.g., phone ringing, cup falling on the floor) rather than the rather artificial distraction of item opacity.

We acknowledge the reviewer's concerns. First, we would like to clarify that in the example highlighted by the reviewer (p.2), we are summarising past research which includes work covering the effects of different forms of distraction. Additionally, we highlight that the distraction manipulation within the current manuscript is related to

visual distraction throughout the manuscript (e.g., title, abstract, description of manipulation in the introduction, methods, discussion, conclusion). Further, we specifically note that referring to “distraction” within the results section implies “visual distraction” (see p. 5). In the limitation section we further acknowledge and discuss the fact that we did not consider other common forms of distraction. Overall, we hope this sufficiently clarifies the nature of the source of distraction investigated within the current manuscript.

2. Thank you for highlighting that the items in the resource pool reappeared after being moved and for presenting the data looking at temporal effects in Supplementary Notes 2. I proposed this as an exploratory analysis and think it is sufficient to just include the figures to allow a visualisation of temporal effects rather than statistics. Given the focus of the paper on naturalistic behaviour, I think the main text would benefit from the inclusion of this analysis which clearly shows temporal effects (although I do acknowledge this may not be possible due to word limits).

We agree with the reviewer that any temporal task effects are an important aspect of naturalistic behaviour and now include this in the Supplementary Information. Additionally, we agree that the exploratory (reviewer-requested) nature of these analyses should be transparent, and we now mark this more clearly throughout the manuscript and Supplementary Information.

For example, on p. 11:

“Given the temporally-extended nature of each trial, we conducted a post-hoc analysis into potential temporal effects in the encoding subcomponent, showing that the effects of distraction and movement effort go above and beyond any temporal task effects (see Supplementary Notes 2).”

3. The conclusions from the section ‘Distraction decreases memory usage’ do not really fit with the introduction to the section ‘Consequences of distraction on sensorimnemonic decisions.’ I acknowledge the authors addition of the word ‘slightly’ but do not agree that Figure 5 supports the conclusion ‘When encountering high distraction, participants use a slightly lower number of attributes in WM’ (Page 14, line 23). The integer number is the same. The authors acknowledge this themselves ‘participants predominantly relied on only one attribute when movement effort was low’ (Page 15, line 11). Here it becomes clear why the movement effort variable was included.

We believe the reviewer is referring to the fact that our conclusion “distraction decreases memory usage” is based on the average number of attributes used in WM, as well as the average probability of using a certain number of attributes in memory, but this is not explicitly mentioned in the introduction to the section on sensorimnemonic decisions. We now make this explicit to avoid any possible confusion (p.12):

“Our findings from the encoding and WM usage metrics converge on a consistent pattern: When encountering high distraction, participants on average use a slightly lower number of attributes in WM. Instead, they encode from the external environment more frequently.”

Further, the reviewer is correct that we do not observe a largescale shift in the overall behavioural strategy in response to distraction (e.g., when movement effort was low, participants predominantly relied on only one attribute). Nevertheless, we present consistent evidence in the metrics of WM usage, encoding, and sensorimnemonic decisions showing that distraction led to a slight change in memory-guided behaviour. That is, participants on average used less memory and encoded from the environment more frequently. This conclusion is not dependent on the inclusion of movement effort, as it emerges in both high and low movement effort.

4. I find some statements are rather vague and would benefit from more specific terminology. For example, ‘how specific perturbations in sensory parameters of the environment can have multiple cascading consequences for processes that are contingent on the affected subcomponent.’ (Page 3, line 16). ‘Perturbations in sensory parameters of the environment’ sounds very elaborate given the actual manipulation is the opacity of the items.

We revised this section to use more accessible terminology (p. 3):

“Second, we illustrate how specific changes in sensory parameters of the environment can have multiple cascading consequences for processes that are contingent on the affected subcomponent.”

5. Thank you for the explanation on Figure 1a. Perhaps it’s helpful to rotate the heading ‘resources’ to be in the participants viewpoint as well and be consistent with the heading ‘model’.

Thank you for your suggestion and feedback on Figure 1a. We played around with different rotations and finally chose the rotation which seemed most clear.